# The mechanosensitive Piezo1 channel is required for bone formation

Weijia Sun[1†], Shaopeng Chi[2†], Yuheng Li[1], Shukuan Ling[1], Yingjun Tan[1], Youjia Xu[3], Fan Jiang[2], Jianwei Li[1], Caizhi Liu[1], Guohui Zhong[1], Dengchao Cao[1], Xiaoyan Jin[1], Dingsheng Zhao[1], Xingcheng Gao[1], Zizhong Liu[1], Bailong Xiao[2*], Yingxian Li[1*]

[1]State Key Laboratory of Space Medicine Fundamentals and Application, China Astronaut Research and Training Center, Beijing, China; [2]State Key Laboratory of Membrane Biology, Tsinghua-Peking Joint Center for Life Sciences, IDG/McGovern Institute for Brain Research, School of Pharmaceutical Sciences, Tsinghua University, Beijing, China; [3]The Second Affiliated Hospital of Soochow University, Suzhou, China

**Abstract** Mechanical load of the skeleton system is essential for the development, growth, and maintenance of bone. However, the molecular mechanism by which mechanical stimuli are converted into osteogenesis and bone formation remains unclear. Here we report that Piezo1, a bona fide mechanotransducer that is critical for various biological processes, plays a critical role in bone formation. Knockout of Piezo1 in osteoblast lineage cells disrupts the osteogenesis of osteoblasts and severely impairs bone structure and strength. Bone loss that is induced by mechanical unloading is blunted in knockout mice. Intriguingly, simulated microgravity treatment reduced the function of osteoblasts by suppressing the expression of Piezo1. Furthermore, osteoporosis patients show reduced expression of Piezo1, which is closely correlated with osteoblast dysfunction. These data collectively suggest that Piezo1 functions as a key mechanotransducer for conferring mechanosensitivity to osteoblasts and determining mechanical-load-dependent bone formation, and represents a novel therapeutic target for treating osteoporosis or mechanical unloading-induced severe bone loss.
DOI: https://doi.org/10.7554/eLife.47454.001

**\*For correspondence:**
xbailong@mail.tsinghua.edu.cn
(BX);
yingxianli@aliyun.com (YL)

[†]These authors contributed equally to this work

**Competing interests:** The authors declare that no competing interests exist.

## Introduction

Bone is the vital organ that constantly responds to and adapts to changes in mechanical loads associated with body weight, movement and gravity (*Iwaniec and Turner, 2016*). Such a mechanical-load-induced remodeling process is determined through the functional interaction between the bone-forming osteoblasts and the bone-absorbing osteoclasts. Perturbation of this remodeling process can lead to the well-documented bone-loss phenomenon that occurs upon mechanical unloading during long-term confinement in bed or spaceflight (*Nagaraja and Risin, 2013*). Impaired osteoblast function might contribute to reduced bone formation (*Carmeliet and Bouillon, 2001*). However, the mechanical response properties and the underlying mechanotransduction molecules that are active during bone formation remain poorly understood.

The mechanosensitive Piezo1 channel (*Coste et al., 2010*; *Coste et al., 2012*; *Ge et al., 2015*; *Zhao et al., 2016*; *Zhao et al., 2018*) mediates mechanical responses in various cell types (*Geng et al., 2017*; *Murthy et al., 2017*), including vascular and lymphatic endothelial cells (*Choi et al., 2019*; *Li et al., 2014*; *Nonomura et al., 2018*; *Ranade et al., 2014*), smooth muscle cells (*Retailleau et al., 2015*), red blood cells (*Cahalan et al., 2015*), epithelial cells (*Eisenhoffer et al., 2012*; *Gudipaty et al., 2017*), neural stem cells (*Pathak et al., 2014*) and

**eLife digest** The bones in our skeletons are constantly exposed to mechanical forces, including those exerted by our muscles and also Earth's gravity. These forces normally help osteoblasts, the cells which build new bone tissue, ensure that bones grow correctly and remain strong. Removing mechanical loads from bones, however, disrupts this process, leading to rapid loss of bone tissue. This is why both astronauts in space (where gravity is much weaker) and bed-ridden patients often go on to develop brittle bones.

To detect and respond to mechanical forces, cells use specialized sensor proteins. One such 'mechanosensor' is a protein called Piezo1, which is found on the surface of many different types of cells in our bodies. It helps cells respond to touch, pressure, or stretching of the surrounding tissue. For example, Piezo1 in nerve cells underpins our sense of touch, while in the cells lining our blood vessels it senses the force exerted by blood flow.

Although osteoblasts clearly respond to mechanical stimuli, exactly how they do so has remained unknown. Sun et al. therefore wanted to find out if Piezo1 also acted as a mechanosensor in osteoblasts, and if so, what role it might play in the loss or formation of bone tissue after changes in the amount of force the bone is exposed to.

Experiments using mouse cells grown in the laboratory revealed that Piezo1 was present in osteoblasts and did indeed help the cells respond to mechanical impact of being poked by a microscopic probe. Mice that had been genetically engineered to remove Piezo1 from their osteoblasts did not grow properly, appearing stunted in adulthood. In these mice, the bones supporting most of the body's weight were also shorter and weaker.

Crucially, putting normal bone cells in a low-gravity simulator – therefore mimicking space flight – or exposing mice to conditions mimicking bed-rest was enough to reduce the level of Piezo1 in osteoblasts. In human patients with osteoporosis, where bones become brittle with age, a decrease in levels of Piezo1 is correlated with increasing bone loss. These results show that Piezo1 is required to make healthy bone tissue, and that its loss is probably involved in the increasing fragility that occurs when mechanical forces applied to bones are reduced.

This work is an important step towards understanding how our bones are built and maintained. In the future, increasing Piezo1 activity within osteoblasts may lead to treatments for bone loss, whether in hospital patients or astronauts.

DOI: https://doi.org/10.7554/eLife.47454.002

chondrocytes (*Lee et al., 2014*; *Rocio Servin-Vences et al., 2017*). Constitutive knockout of Piezo1 results in embryonic lethality in mice, mainly because of defects in vascular development (*Li et al., 2014*; *Ranade et al., 2014*). Piezo1 senses shear stress in vascular endothelial cells and red blood cells, contributing to the regulation of blood pressure (*Rode et al., 2017*; *Wang et al., 2016*) and of red blood cell volume (*Cahalan et al., 2015*), respectively. Piezo1 can also sense the local cellular environment and thus has a role in epithelial cell homeostasis (*Eisenhoffer et al., 2012*; *Gudipaty et al., 2017*), the lineage choice of neural stem cells (*Pathak et al., 2014*), axon growth (*Koser et al., 2016*) and axon regeneration (*Song et al., 2019*).

Given its widespread function in various cell types and biological processes (*Murthy et al., 2017*), we have reasoned that Piezo1 might play important roles in mechanical-load-dependent bone formation and remodeling. In line with this, expression data obtained from the BioGPS database (http://biogps.org/#goto=genereport&id=234839) indicate that Piezo1 is highly expressed in osteoblasts. Furthermore, a previous study has shown that the expression level of Piezo1 in mesenchymal stem cells (MSCs) was correlated to the promotion of osteoblast differentiation and to the inhibition of adipocyte differentiation (*Sugimoto et al., 2017*). Although Piezo1 was proposed to function as a mechanoreceptor that initiates the response to hydrostatic pressure (HP) in MSCs, the channel activities and the in vivo contribution to bone formation of Piezo1 expressed in MSCs and MSC-derived cell lineages such as osteoblasts have not been directly examined (*Sugimoto et al., 2017*). In the present study, we set out to address whether Piezo1 might function as a key mechanotransducer that confers mechanosensitivity to the bone-forming osteoblasts and that consequently determines mechanical-load-induced bone formation and remodeling. Toward this goal, we systematically

examined the role of Piezo1 in bone formation using Ocn-Cre-dependent Piezo1 knockout mice, mechanical-unloading-induced mouse and cellular models, and human osteoporosis samples. We have found that Piezo1 plays a key role in determining the mechanical response of osteoblasts and the in vivo formation and remodeling of bone in mice and humans.

## Results

### Expression and function of Piezo1 in the osteoblast cell line

To explore the role of Piezo1 in bone cell types, we initially examined the mechanical response of the commonly used pre-osteoblast cell line MC3T3-E1 and the pre-osteoclast cell line RAW264.7. To this end, we directly measured the mechanically evoked cationic currents using whole-cell patch clamp electrophysiology coupled with mechanical poking of the cell membrane with a piezo-driven blunt glass pipette. MC3T3-E1 displayed mechanically activated currents in a step-dependent manner with a maximal current of 87.7 ± 11.6 pA (*Figure 1a,b*). By contrast, much smaller mechanically activated currents were recorded in cell of the pre-osteoclast cell line RAW264.7 (24.0 ± 5.0 pA) (*Figure 1a,b*). In line with the recorded mechanically activated currents, the mRNA of Piezo1 was significantly higher in MC3T3-E1 cells than in RAW264.7 (*Figure 1c*). Furthermore, either siRNA-mediated knockdown of Piezo1 (*Figure 1f,g*) or the application of GsMTX4, a relatively specific blocker of the Piezo channel family (*Bae et al., 2011*), significantly reduced the mechanically activated currents in MC3T3-E1 cells (from 68.6 ± 7.0 pA to 17.3 ± 2.9 pA or 21.7 ± 5.6 pA, respectively) (*Figure 1d,e*). These data suggest that Piezo1 is expressed and mediates the mechanically activated currents in MC3T3-E1 cells. Interestingly, siRNA-mediated knockdown of Piezo1 in MC3T3-E1 cells decreased the expression of the functional marker genes of osteoblasts revealed by QRT-PCR, including alkaline phosphatase (*Alp*), osteocalcin (*Bglap*) and collagen 1 (*Col1α1*) (*Figure 1h*), and reduced the Alp staining (*Figure 1i*). Furthermore, we found that the expression of Piezo1 was increased in MC3T3-E1 cells cultured with osteogenic medium for 1 day, 3 days and 5 days (*Figure 1—figure supplement 1a,b*). These data suggest that Piezo1 might mediate the mechanical response and function of differentiated osteoblasts. Thus, we next focused on characterizing the role of Piezo1 in primary osteoblasts and in vivo bone formation.

### Expression and function of Piezo1 in primary mouse osteoblasts

QRT-PCR analysis revealed that Piezo1 was highly expressed in the bone among the various examined mouse tissues (*Figure 2a*). To investigate the specific expression and function of Piezo1 in the bone-forming osteoblasts, we chose Ocn-Cre mice—which specifically express Cre recombinase under the osteocalcin (Ocn) gene promoter in osteoblast-lineage cells, but not in osteoclasts (*Huang et al., 2016*; *Zhang et al., 2002*)—to generate the conditional Piezo1 KO mice (Piezo1$^{Ocn/Ocn}$) by crossing the Ocn-Cre mice with the Piezo1$^{fl/fl}$ mice (*Cahalan et al., 2015*). In line with the study of MC3T3-E1 cells (*Figure 1*), we detected mRNA and protein expression of Piezo1 in the bone tissue and primary osteoblasts derived from littermate Piezo1$^{fl/fl}$ mice (*Figure 2b–e*). The Piezo1$^{Ocn/Ocn}$ mice show significantly reduced expression of Piezo1 specifically in the bone, but not other tissues (*Figure 2b,c*), consistent with the specific knockout of Piezo1 in the bone. Furthermore, osteoblasts that were derived from Piezo1$^{Ocn/Ocn}$ mice also had drastically reduced expression of Piezo1 (*Figure 2d,e*). Notably, the reduction of Piezo1 was more complete in the osteoblasts than in the whole bone tissue, indicating the possible expression of Piezo1 in non-osteoblast-lineage cells (*Figure 2b–e*), which could not be ablated by using Ocn-Cre mice. We quantitatively compared the mechanically evoked currents of primary osteoblasts isolated from Piezo1$^{fl/fl}$ and Piezo1$^{Ocn/Ocn}$ mice. Consistent with the recordings from MC3T3-E1 cells, Piezo1$^{fl/fl}$ osteoblasts displayed poking-induced currents in a step-dependent manner with a maximal current of 152.3 ± 21.6 pA (*Figure 2f, g*). The inactivation Tau of the current is 32.8 ± 4.1 ms (*Figure 2f,h*). The maximal currents of the Piezo1$^{Ocn/Ocn}$ osteoblasts were significantly reduced to 66.0 ± 14.8 pA (*Figure 2f,g*), while the inactivation Tau was unchanged (27.8 ± 3.9 ms) (*Figure 2f,h*). The remaining mechanically evoked currents could be due to incomplete knockout of Piezo1, as residual Piezo1 proteins were detected in the Piezo1$^{Ocn/Ocn}$ osteoblasts (*Figure 2e*) and the excision rate in osteoblasts of Ocn-Cre mice was estimated to be ~88% (*Zhang et al., 2002*). Alternatively, other mechanosensitive channels that are independent of Piezo1 might account for the remaining mechanically activated currents in

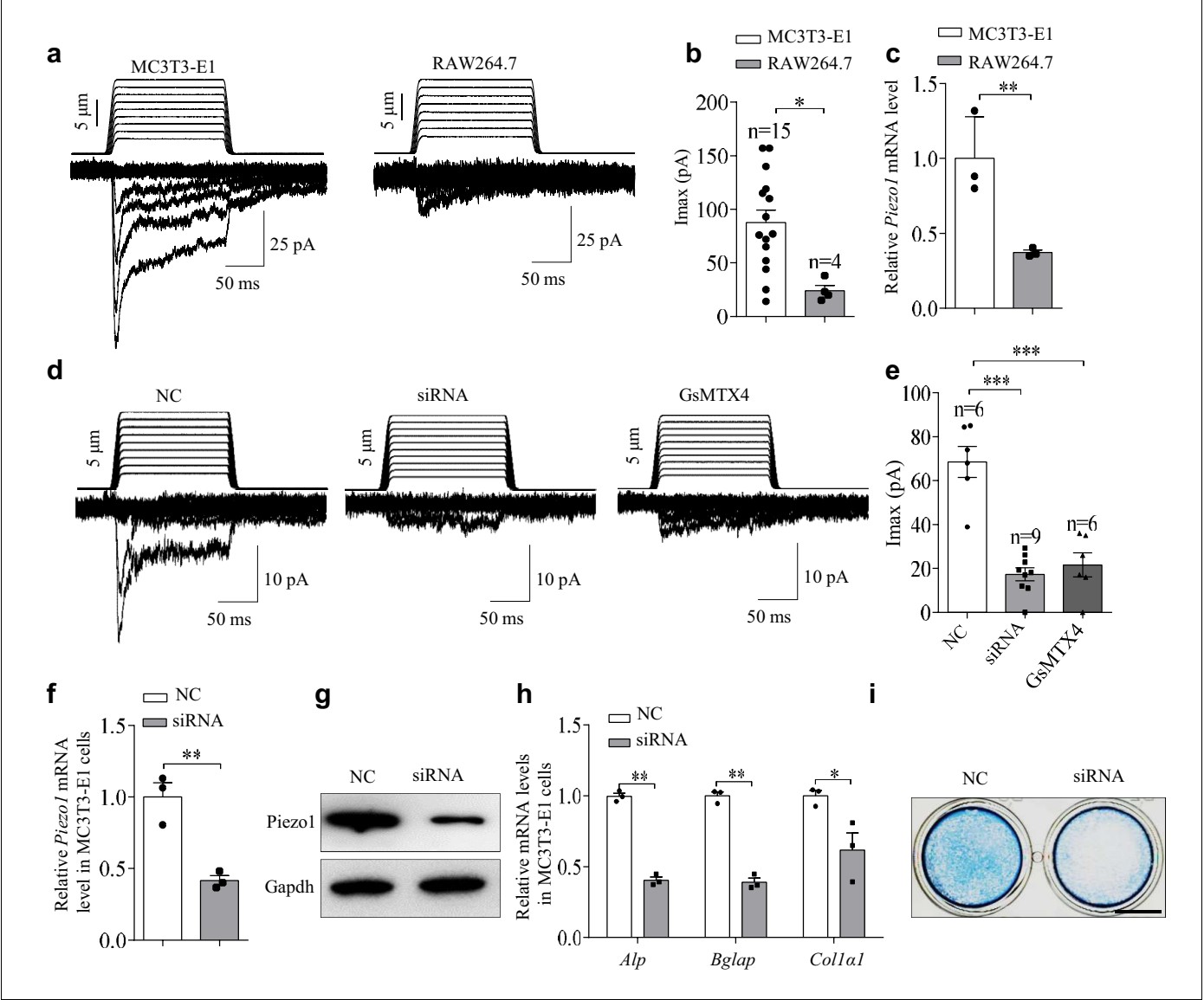

**Figure 1.** The expression and function of Piezo1 in the pre-osteoblast cell line MC3T3-E1. (a) Representative traces of poking-induced inward currents recorded at −60 mV in the indicated cell lines. (b) Scatter plots of the maximal poking-induced currents in the indicated cell lines. (c) QRT-PCR analysis of *Piezo1* mRNA level in the indicated cell lines. (d) Representative traces of poking-induced inward currents recorded at −60 mV in MC3T3-E1 under the indicated conditions. 'siRNA' indicates the siRNA-mediated knockdown of Piezo1. NC, Piezo1 negative-control siRNA. GsMTX4 is a relatively specific blocker of the Piezo channel family. (e) Scatter plots of the maximal poking-induced currents in MC3T3-E1 cells under the indicated conditions. (f, g) QRT-PCR analysis of Piezo1 mRNA level (f) and western blot analysis of Piezo1 protein level (g) in MC3T3-E1 cells transfected with control or Piezo1 siRNA for 48 hr. (h) QRT-PCR analysis of *Alp*, *Bglap*, and *Col1α1* mRNA levels in MC3T3-E1 cells transfected with control or *Piezo1* siRNA for 48 hr. (i) Representative images of Alp staining in MC3T3-E1 cells transfected with control or Piezo1 siRNA for 48 hr. Scale bar, 5 mm. *, p<0.05; **, p<0.01; ***, p<0.001.

DOI: https://doi.org/10.7554/eLife.47454.003

The following source data and figure supplement are available for figure 1:

**Source data 1.** The expression and function of Piezo1 in pre-osteoblast cell line MC3T3-E1.
DOI: https://doi.org/10.7554/eLife.47454.005
**Figure supplement 1.** Osteogenic-induced expression of Piezo1 in the pre-osteoblast cell line MC3T3-E1.
DOI: https://doi.org/10.7554/eLife.47454.004

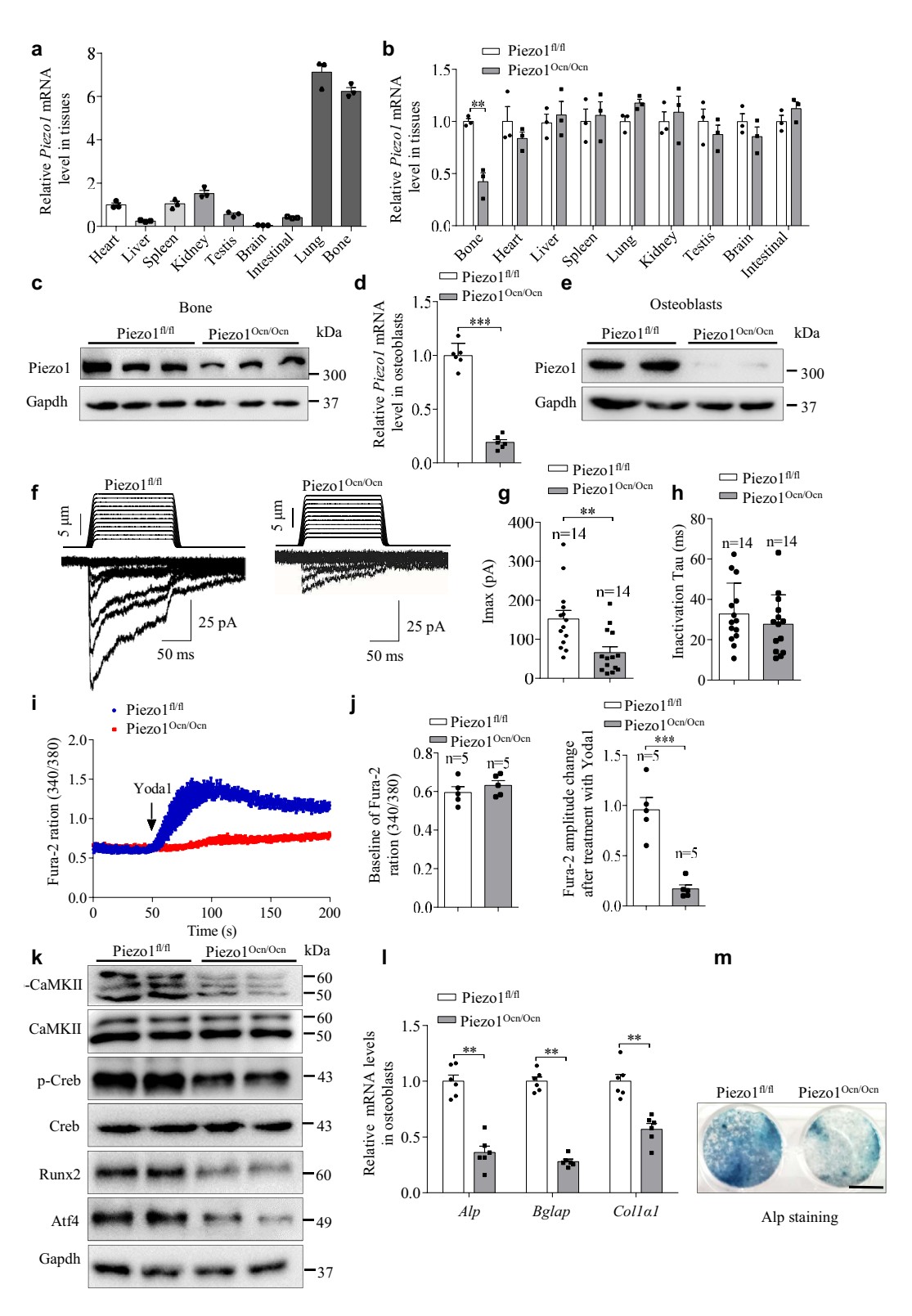

**Figure 2.** Expression and function of Piezo1 in primary osteoblasts isolated from Piezo1[fl/fl] and Piezo1[Ocn/Ocn] mice. (a) QRT-PCR analysis of *Piezo1* mRNA level in bone and other tissues from WT mice. (b) QRT-PCR analysis of *Piezo1* mRNA level in different tissues from Piezo1[fl/fl] or Piezo1[Ocn/Ocn] mice. The *Piezo1* mRNA level in all tissues from Piezo1[Ocn/Ocn] mice was normalized to that in Piezo1[fl/fl] mice. (c) Western blot analysis of Piezo1 protein level in bone tissues from Piezo1[fl/fl] and Piezo1[Ocn/Ocn] mice. (d) QRT-PCR analysis of *Piezo1* mRNA level in primary osteoblasts isolated from Piezo1[fl/fl]

*Figure 2 continued on next page*

*Figure 2 continued*

or Piezo1$^{Ocn/Ocn}$ mice and cultured with osteogenic medium for 5 days. (e) Western blot analysis of Piezo1 protein level in primary osteoblasts cultured with osteogenic medium for 5 days. (f) Representative traces of poking-induced inward currents recorded at −60 mV in primary osteoblasts isolated from Piezo1$^{fl/fl}$ or Piezo1$^{Ocn/Ocn}$ mice. (g, h) Scatter plots of the maximal poking-induced currents (g) and inactivation tau (h) in primary osteoblasts isolated from Piezo1$^{fl/fl}$ or Piezo1$^{Ocn/Ocn}$ mice. (i) Average single-cell Ca$^{2+}$ imaging traces of Piezo1$^{fl/fl}$ or Piezo1$^{Ocn/Ocn}$ osteoblasts showing the 340/380 ratio of the Ca$^{2+}$-sensitive Fura-2 dye in response to the application of 30 µM Yoda1. (j) Scatter plot of baseline and Yoda1-induced Fura-2 amplitude changes in Piezo1$^{fl/fl}$ or Piezo1$^{Ocn/Ocn}$ osteoblasts. (k) Western blot analysis of p-CaMKII, CaMKII, p-Creb, Creb, Runx2 and Atf4 proteins in primary osteoblasts cultured with osteogenic medium for 5 days. (l) QRT-PCR analysis of *Alp*, *Bglap* and *Col1α1* mRNA levels in primary osteoblasts cultured with osteogenic medium for 5 days. (m) Representative images of Alp staining in primary osteoblasts cultured with osteogenic medium for 5 days. Scale bar, 5 mm. The staining data were confirmed by three repeated tests. All data are the mean ± s.e.m. from three independent experiments. **, $p < 0.01$; ***, $p < 0.001$.

DOI: https://doi.org/10.7554/eLife.47454.006

The following source data and figure supplements are available for figure 2:

**Source data 1.** The expression and function of Piezo1 in primary osteoblasts isolated from Piezo1$^{fl/fl}$ and Piezo1$^{Ocn/Ocn}$ mice.
DOI: https://doi.org/10.7554/eLife.47454.009

**Figure supplement 1.** Characterizations of primary osteoblasts derived from Piezo1fl/fl mice with or without ectopic expression of the Cre recombinase.
DOI: https://doi.org/10.7554/eLife.47454.007

**Figure supplement 2.** Expression and function of Piezo1 in osteocytes isolated from Piezo1$^{fl/fl}$ and Piezo1$^{Ocn/Ocn}$ mice.
DOI: https://doi.org/10.7554/eLife.47454.008

the Piezo1$^{Ocn/Ocn}$ osteoblasts. Nevertheless, these data demonstrate that Piezo1 is expressed and functionally mediates mechanically evoked responses in osteoblasts.

Piezo1 is a non-selective cation channel that allows Ca$^{2+}$ influx and initiation of downstream Ca$^{2+}$ signaling events upon its opening. We therefore assayed whether Piezo1 mediates Ca$^{2+}$ influx in osteoblasts. Using single-cell Ca$^{2+}$ imaging with the ratiometric Ca$^{2+}$ dye Fura2, we found that Yoda1, a previously identified Piezo1 chemical activator (*Syeda et al., 2015*), induced a Ca$^{2+}$ response in WT osteoblasts, which was drastically reduced in Piezo1$^{Ocn/Ocn}$ osteoblasts (*Figure 2i,j*). Previous studies have shown that Ca$^{2+}$ influx could lead to phosphorylation of CaMKII and activate the Creb pathway, promoting osteoblast differentiation (*Choi et al., 2013*; *Zayzafoon et al., 2005*). A Piezo-dependent Ca$^{2+}$-CaMKII signaling pathway has been reported in the axon regeneration process (*Song et al., 2019*). In line with these previous findings, we found that the phosphorylation of CaMKII and Creb was apparently reduced in osteoblasts derived from the Piezo1$^{Ocn/Ocn}$ cells (*Figure 2k*). Furthermore, Runx2 and Atf4, key transcription factors involved in osteoblast differentiation, were downregulated in the Piezo1$^{Ocn/Ocn}$ cells (*Figure 2k*). Consistently, the Piezo1$^{Ocn/Ocn}$ osteoblasts showed decreased expression of the differentiation marker genes (*Figure 2l*) and reduced Alp activity (*Figure 2m*), indicating impaired osteogenesis.

To investigate whether Piezo1 plays an autonomous role in osteoblasts instead of in osteoblast precursors, primarily cultured osteoblasts derived from Piezo1$^{fl/fl}$ mice were transfected with the pIRES-EGFP control plasmid (Ctrl) or the pCAG-Cre-IRES2-GFP (Cre) plasmid to delete Piezo1. The expression level of Piezo1 in cells transfected with the Cre plasmid was reduced to ~60% of that in cells transfected with the Ctrl plasmid, which is in line with the transfection efficiency (*Figure 2— figure supplement 1a,b*). The maximal current of cells transfected with the control plasmid was 102.6 ± 13.97 pA, whereas in the Piezo1$^{fl/fl}$ osteoblasts transfected with the Cre plasmid the maximum current was reduced to 32.72 ± 5.7 pA (*Figure 2—figure supplement 1c,d*). Consistent with the results obtained from the Piezo1$^{Ocn/Ocn}$ osteoblasts, the phosphorylation levels of CaMKII and Creb, as well as the levels of Runx2 and Atf4, were reduced in the Piezo1$^{fl/fl}$ osteoblasts that were transfected with the Cre plasmid (*Figure 2—figure supplement 1e*). Furthermore, the Piezo1$^{fl/fl}$ osteoblasts that were transfected with the Cre plasmid showed decreased expression of the differentiation marker genes (*Figure 2—figure supplement 1f*), as well as reduced Alp activity and mineral deposition (*Figure 2—figure supplement 1g,h*).

Given that Ocn-Cre mice express the Cre recombinase under the osteocalcin (Ocn) gene promoter in osteoblast-lineage cells, we also examined the expression and function of Piezo1 in primary osteocytes isolated from the Piezo1$^{fl/fl}$ and Piezo1$^{Ocn/Ocn}$ mice. Indeed, osteocytes derived from the Piezo1$^{Ocn/Ocn}$ mice exhibited a reduced level of Piezo1 (*Figure 2—figure supplement 2a,*

*b*). Piezo1$^{fl/fl}$ osteocytes displayed poking-induced currents in a step-dependent manner with a maximal current of 64.9 ± 13.7 pA, which was significantly reduced to 34.2 ± 4.1 pA in the Piezo1$^{Ocn/Ocn}$ osteocytes (*Figure 2—figure supplement 2c,d*). It has been shown that osteocytes were able to coordinate osteogenesis in response to mechanical stimulation through the regulation of Sost level (*Robling et al., 2008*). We found that *Sost* expression was upregulated in bone tissue derived from Piezo1$^{Ocn/Ocn}$ mice compared to that from Piezo1$^{fl/fl}$ mice, which is consistent with the reduced osteoblast function observed in Piezo1$^{Ocn/Ocn}$ mice.

Collectively, these data suggest that Piezo1 functions as a critical mechanotransduction channel in osteoblast-derived lineage cells in bone, including both osteoblasts and osteocytes.

## The Piezo1$^{Ocn/Ocn}$ mice show defective bone formation

We next examined the in vivo role of Piezo1 in bone formation. Alizarin red and Alcian blue staining revealed that the newborn Piezo1$^{Ocn/Ocn}$ mice had skeletal size similar to that of their Piezo1$^{fl/fl}$ littermates (*Figure 3a*). However, the Piezo1$^{Ocn/Ocn}$ mice exhibited incomplete closure of the cranial structure (*Figure 3a*). At 8 weeks of age, the male Piezo1$^{Ocn/Ocn}$ mice showed shorter stature (*Figure 3—figure supplement 1a*) and lower body weight (*Figure 3—figure supplement 1b*). The length of the femur and tibia of the Piezo1$^{Ocn/Ocn}$ mice was apparently shorter than that of the Piezo1$^{fl/fl}$ control mice (*Figure 3—figure supplement 1c*).

We next carried out micro-CT analysis. The bodyweight-bearing long bones of the Piezo1$^{Ocn/Ocn}$ mice exhibited drastic loss of bone mass, reduced thickness and impaired trabeculation (*Figure 3b, c*). In the Piezo1$^{Ocn/Ocn}$ mice, bone parameters including the trabecular bone mineral density (BMD), bone volume (BV/TV), trabecular number (Tb.N), trabecular thickness (Tb.Th) and cortical bone thickness (Cort.Th) were all significantly decreased, whereas the trabecular spacing (Tb.Sp) was accordingly increased (*Figure 3d*). The strength of the long bones of the Piezo1$^{Ocn/Ocn}$ mice was only about half of that of the Piezo1$^{fl/fl}$ control littermates (*Figure 3e*). Moreover, the rate of bone formation and bone formation rate per area of bone surface were significantly reduced in the Piezo1$^{Ocn/Ocn}$ mice (*Figure 3f*). Consistent with the bone defects, immunostaining analysis revealed that the expression levels of the osteoblast differentiation markers, including Col1α1 and Ocn, were reduced in the tibias of the Piezo1$^{Ocn/Ocn}$ mice (*Figure 3g*). Accordingly, the levels of Ocn and PINP (N-Propeptide of type I Procollagen) were significantly decreased in serum derived from the Piezo1$^{Ocn/Ocn}$ mice (*Figure 3h*). Furthermore, the expression of the osteoblast differentiation marker genes was significantly decreased in femurs isolated from the Piezo1$^{Ocn/Ocn}$ mice (*Figure 3i*). However, the osteoclast activity remained similar in the Piezo1$^{fl/fl}$ and Piezo1$^{Ocn/Ocn}$ mice (*Figure 3—figure supplement 1d–f*), suggesting an osteoblast-lineage-specific defect in bone formation in the Piezo1$^{Ocn/Ocn}$ mice. As shown in *Figure 4*, the Piezo1$^{Ocn/Ocn}$ mice examined at 16 weeks of age showed bone defects that were essentially similar to those of mice at 8 weeks of age (*Figure 3*).

We also analyzed the in vivo role of Piezo1 in female mice. In the Piezo1$^{Ocn/Ocn}$ female mice, the expression of Piezo1 was reduced (*Figure 3—figure supplement 2a,b*). The bone parameters including the BMD, BV/TV, Tb.N, Tb.Th and Cort.Th were all significantly decreased, whereas the Tb.Sp was accordingly increased (*Figure 3—figure supplement 2c,d*). The expression of the osteoblast differentiation marker genes was significantly reduced in femurs isolated from the Piezo1$^{Ocn/Ocn}$ female mice (*Figure 3—figure supplement 2e*). Thus, the Piezo1$^{Ocn/Ocn}$ female mice showed a phenotype similar to that of male mice. Taken together, these data suggest that Piezo1 deficiency in osteoblast-lineage cells significantly impairs the formation and structural integrity of the bone.

## The role of Piezo1 in mechanical-unloading-induced bone loss

To determine whether a Piezo1-mediated mechanical response in the bone is responsible for mechanical-unloading-induced bone loss, we employed the commonly used hindlimb suspension (HS) model (*Wang et al., 2013*; *Xu et al., 2017*) to examine the bone remodeling process in response to the weight-bearing unloading of the Piezo1$^{fl/fl}$ and Piezo1$^{Ocn/Ocn}$ mice. When the Piezo1$^{fl/fl}$ mice were subjected to 28 days of HS, the trabecular bone mass and architecture related parameters, including BMD, BV/TV, Tb.N and Tb.Th, were significantly reduced (*Figure 4a,b*). These HS-induced phenotypes of the Piezo1$^{fl/fl}$ mice essentially resemble the bone deficits observed in the Piezo1$^{Ocn/Ocn}$ mice without HS treatment (*Figure 4a,b*). In contrast to the Piezo1$^{fl/fl}$ mice, the Piezo1$^{Ocn/Ocn}$ mice that were subjected to the HS treatment did not show a worsened phenotype in its

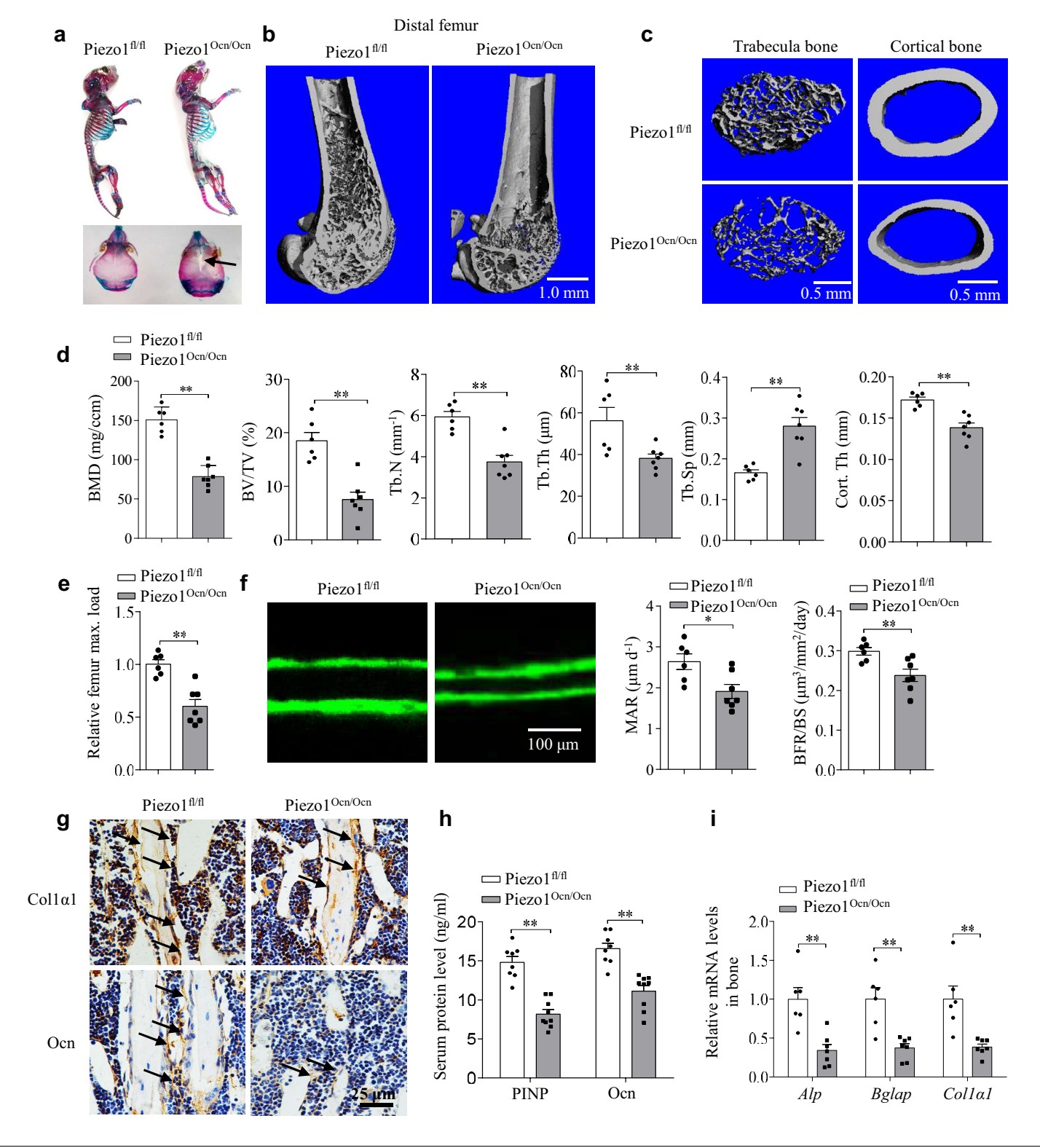

**Figure 3.** Piezo1[Ocn/Ocn] mice show severely impaired bone formation. (**a**) Alcian blue and Alizarin red staining of the skeletons of Piezo1[fl/fl] or Piezo1[Ocn/Ocn] mice at 7 days old. (**b, c**) Representative images showing the three-dimensional cortical bone and trabecular architecture as shown by micro-CT reconstruction at the distal femurs from Piezo1[fl/fl] or Piezo1[Ocn/Ocn] mice at 2 months old. Scale bars: (**b**) 1.0 mm, (**c**) 0.5 mm. (**d**) Micro-CT measurements for bone mineral density (BMD), trabecular bone volume fraction (BV/TV), trabecular number (Tb.N), trabecular thickness (Tb.Th), trabecular separation (Tb.Sp) and cortical thickness (Cort.Th) at the distal femurs from Piezo1[fl/fl] (n = 6) or Piezo1[Ocn/Ocn] (n = 7) mice. (**e**) Relative maximal (max.)

*Figure 3 continued on next page*

*Figure 3 continued*

load at failure determined by three-point bending of femurs from Piezo1$^{fl/fl}$ (n = 6) and Piezo1$^{Ocn/Ocn}$ mice (n = 7). (**f**) Representative images showing new bone formation assessed by double calcein labeling in Piezo1$^{fl/fl}$ mice (n = 6) and Piezo1$^{Ocn/Ocn}$ mice (n = 7). MAR, mineral apposition rate. BFR/BS, bone formation rate/bone surface. Scale bar, 100 µm. (**g**) Histology images for Col1α1 and Ocn staining of the proximal tibia from Piezo1$^{fl/fl}$ and Piezo1$^{Ocn/Ocn}$ mice. Scale bar, 25 µm. (**h**) ELISA analysis of the levels of PINP and Ocn protein levels in the serum from Piezo1$^{fl/fl}$ (n = 8) and Piezo1$^{Ocn/Ocn}$ mice (n = 9). (**i**) QRT-PCR analysis of *Alp*, *Bglap* and *Col1α1* mRNA levels in bone tissues collected from Piezo1$^{fl/fl}$ (n = 6) and Piezo1$^{Ocn/Ocn}$ mice (n = 7). All data are the mean ± s.e.m. *, p<0.05; **, p<0.01.

DOI: https://doi.org/10.7554/eLife.47454.010

The following source data and figure supplements are available for figure 3:

**Source data 1.** The data and statistical analysis of Piezo1$^{Ocn/Ocn}$ mice show severely impaired bone formation.

DOI: https://doi.org/10.7554/eLife.47454.013

**Figure supplement 1.** The phenotypes of Piezo1$^{fl/fl}$ and Piezo1$^{Ocn/Ocn}$ mice.

DOI: https://doi.org/10.7554/eLife.47454.011

**Figure supplement 2.** The phenotypes of female Piezo1$^{fl/fl}$ and Piezo1$^{Ocn/Ocn}$ mice.

DOI: https://doi.org/10.7554/eLife.47454.012

already impaired bone (*Figure 4a,b*). Furthermore, HS treatment led to a significant reduction in the bone strength of the hindlimb of the Piezo1$^{fl/fl}$ mice, but not the Piezo1$^{Ocn/Ocn}$ mice (*Figure 4c*). We observed the corresponding change in the osteoblast function. HS-induced reduction of Ocn and Col1α1 staining in bone tissues (*Figure 4d*), serum PINP and Ocn levels (*Figure 4e*), as well as the expression of the differentiation marker genes of osteoblasts (*Figure 4f*), were specifically observed in the Piezo1$^{fl/fl}$ mice but not in the Piezo1$^{Ocn/Ocn}$ mice. These results demonstrate that the Piezo1$^{fl/fl}$ mice show drastic mechanical-unloading-induced remodeling of the bone, whereas the Piezo1$^{Ocn/Ocn}$ mice are essentially resistant to such remodeling, suggesting that Piezo1 functions as a critical mechanotransducer for the mediation of proper mechanical-load-induced bone remodeling.

## HS or simulated microgravity treatment suppresses the expression of Piezo1 and impairs osteoblast function

Given the observation that the HS-treated WT mice essentially recapitulated the defective bone phenotypes and osteoblast dysfunction of the Piezo1$^{Ocn/Ocn}$ mice, we asked whether mechanical unloading might lead to decreased expression of Piezo1 in the bone tissue derived from the Piezo1$^{Ocn/Ocn}$ mice. Indeed, the mRNA and protein levels of Piezo1 were significantly reduced in the Piezo1$^{fl/fl}$ mice, but there was no further reduction in these levels in the Piezo1$^{Ocn/Ocn}$ mice after treatment with HS for 28 days (*Figure 5a,b*). To examine whether mechanical unloading directly alters Piezo1 expression in osteoblasts, we utilized a cell rotation system to generate a microgravity condition that simulated the effect of mechanical unloading on osteoblasts. Intriguingly, when subjected to the simulated microgravity treatment, the primarily derived osteoblasts had significantly reduced expression of Piezo1 (*Figure 5c,d*). Furthermore, microgravity-treated osteoblasts showed decreased mechanically activated currents (*Figure 5e,f*). In line with the importance of Piezo1 in determining osteoblast activities, the simulated microgravity treatment led to significantly reduced expression of osteoblast marker genes (*Figure 5g*) and Alp activity (*Figure 5h*) in the Piezo1$^{fl/fl}$ cells, but not in the Piezo1$^{Ocn/Ocn}$ cells. These data suggest that mechanical unloading can affect the expression of Piezo1, resulting in dysfunction of osteoblasts and bone formation.

## Mechanical-loading treatment promotes the expression of Piezo1 and osteoblast function

To determine whether Piezo1-mediated mechanical response in the bone is responsible for mechanical-loading-induced bone formation, we subjected the Piezo1$^{fl/fl}$ and Piezo1$^{Ocn/Ocn}$ mice to an exercise model involving a treadmill for 21 days (*Wallace et al., 2007*). The mRNA and protein levels of Piezo1 were significantly increased in the Piezo1$^{fl/fl}$ mice, but not in the Piezo1$^{Ocn/Ocn}$ mice (*Figure 6a,b*). Exercise treatment led to a significant increase in the expression of the osteoblast function marker genes in the Piezo1$^{fl/fl}$ mice, but not in the Piezo1$^{Ocn/Ocn}$ mice (*Figure 6c*). These data suggest that Piezo1 might respond to mechanical loading in a way that affects bone function.

To further examine whether mechanical force directly alters Piezo1 expression in osteoblasts, we utilized fluid shear stress (FSS) to stimulate osteoblasts. When subjected to 12 dyn/cm$^2$ FSS

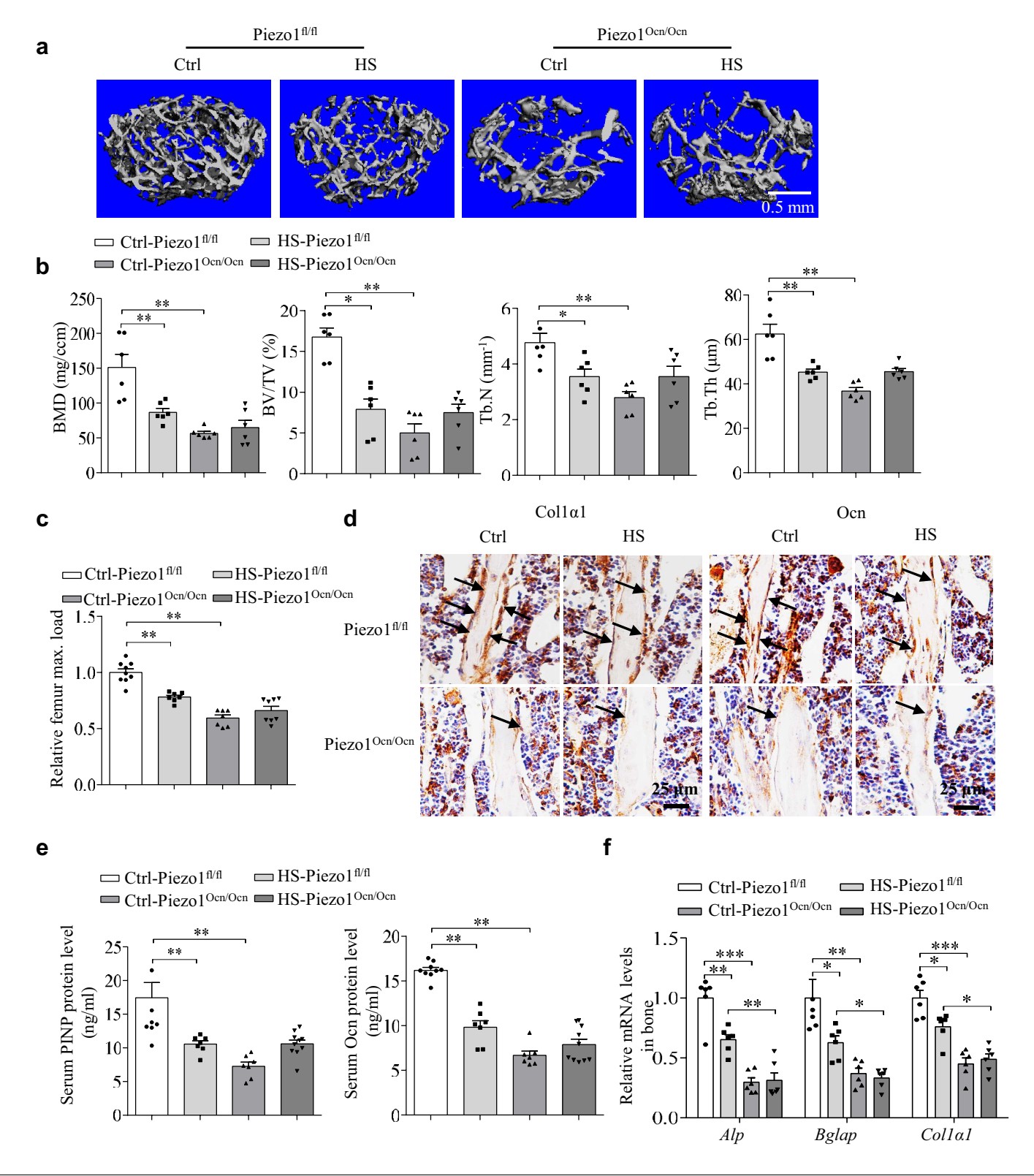

**Figure 4.** The effect of mechanical unloading on bone remodeling and osteoblast function in Piezo1fl/fl and Piezo1Ocn/Ocn mice. (**a**) Representative images showing three-dimensional trabecular architecture as determined by micro-CT reconstruction of the distal femurs from the groups of mice indicated. Scale bar, 0.5 mm. (**b**) Micro-CT measurements for BMD, BV/TV, Tb.N and Tb.Th in the distal femurs from the groups of mice indicated.

*Figure 4 continued on next page*

*Figure 4 continued*

n = 6 in each group. (c) Relative maximal (max.) load at failure determined by three-point bending of femurs from the groups of mice indicated. Ctrl-Piezo1$^{fl/fl}$ group, n = 9; HS-Piezo1$^{fl/fl}$ group, n = 7; Ctrl- Piezo1$^{Ocn/Ocn}$ group, n = 7; HS-Piezo1$^{Ocn/Ocn}$ group, n = 8. (d) Histology images for Col1α1 and Ocn staining of the proximal tibia from the groups of mice indicated. Scale bar: 25 μm. (e) ELISA analysis of the levels of PINP and Ocn proteins in serum from the groups of mice indicated. Ctrl- Piezo1$^{fl/fl}$ group, n = 9; HS-Piezo1$^{fl/fl}$ group, n = 7; Ctrl-Piezo1$^{Ocn/Ocn}$ group, n = 7; HS-Piezo1$^{Ocn/Ocn}$ group, n = 10. (f) QRT-PCR analysis of *Alp*, *Bglap* and *Col1α1* mRNA levels in bone tissues collected from the groups of mice indicated. n = 6 in each group. All data are the mean ± s.e.m. *, p<0.05; **, p<0.01; ***, p<0.001.

DOI: https://doi.org/10.7554/eLife.47454.014

The following source data is available for figure 4:

**Source data 1.** The data and statistical analysis of the effect of mechanical unloading on bone remodeling and osteoblast function in Piezo1$^{fl/fl}$ and Piezo1$^{Ocn/Ocn}$ mice.

DOI: https://doi.org/10.7554/eLife.47454.015

treatment for 2 hr, the primary osteoblasts derived from Piezo1$^{fl/fl}$ mice had significantly increased expression of Piezo1 (*Figure 6d,e*) and of the osteoblast marker genes (*Figure 6f*), as well as increased Alp activity (*Figure 6g*). By contrast, the FSS-induced effect was not observed in the Piezo1$^{Ocn/Ocn}$ cells. These data suggest that Piezo1 can sense mechanical force and can consequently regulate its own expression, osteoblast function and bone formation.

## Decreased Piezo1 expression correlates with defective osteoblast function in osteoporosis patients and mouse models

The close relationship between PIEZO1 expression and function in osteoblasts and bone formation prompted us to explore the pathological role of Piezo1 in human osteoporosis. We examined the expression of Piezo1 in femurs from patients with fractures (*Supplementary file 1*). Interestingly, the mRNA and protein expression levels of Piezo1 in osteoporosis patients (T ≤ −2.5) were significantly lower than those in control patients (T > −2.5) (*Figure 7a,b*). Furthermore, the expression of Piezo1 was positively correlated with the expression of the differential marker genes of osteoblasts, including *ALP*, *BGLAP* and *COL1A1*, in these human samples (*Figure 7c*). By contrast, there are no correlations between the expression of Piezo1 and that of the osteoclast marker genes, including Cathepsin K (*CTSK*), Tartrate-resistant acid phosphatase (*ACP5*) and Matrix metalloproteinase 9 (*MMP9*) (*Figure 7d*). However, there are also no correlations between the expression of Piezo1 and that of the osteocyte marker genes, including Dentin matrix acidic phosphoprotein 1 (*DMP1*) and Sclerostin (*SOST*) (*Figure 7e*). These data are consistent with the critical role of Piezo1 in osteogenesis and bone formation observed in mouse models.

## Discussion

Bone undertakes a life-time mechanical-loading-induced remodeling process. However, it remains unclear how the bone tissue directly senses mechanical changes. In the present study, we have focused on characterizing the expression and function of the mechanically activated Piezo1 channel in bone-forming osteoblasts. Piezo1 is characteristically activated by various forms of mechanical stimulation, including poking, stretching, shear stress and substrate deflection (*Coste et al., 2010*; *Cox et al., 2016*; *Lewis and Grandl, 2015*; *Poole et al., 2014*), and mediates Ca$^{2+}$ influx upon opening to initiate downstream Ca$^{2+}$ signaling, such as the activation of Ca$^{2+}$-dependent calpain (*Li et al., 2014*), eNOS (*Wang et al., 2016*) and CaMKII (*Coste et al., 2010*; *Cox et al., 2016*; *Lewis and Grandl, 2015*; *Song et al., 2019*). Piezo1 was directly activated by asymmetric membrane curvature (*Syeda et al., 2016*) and lateral membrane tension (*Cox et al., 2016*; *Lewis and Grandl, 2015*). In line with being an exquisite mechanosensitive cation channel, the full-length 2547-residue mouse Piezo1 forms a remarkable homo-trimeric three-bladed, propeller-like architecture, comprising a central ion-conducting pore module and three highly curved and non-planar blades, each of which is composed of nine transmembrane helical units (THUs) each of four transmembrane helices (TM) that are connected to the pore by a 9 nm-long beam-like structure (*Ge et al., 2015*; *Guo and MacKinnon, 2017*; *Saotome et al., 2018*; *Zhao et al., 2018*). The blade-beam structure might form a lever-like intramolecular transduction pathway for long-distance mechanical gating of the central pore (*Wang et al., 2018*; *Zhao et al., 2018*). Bone-residing

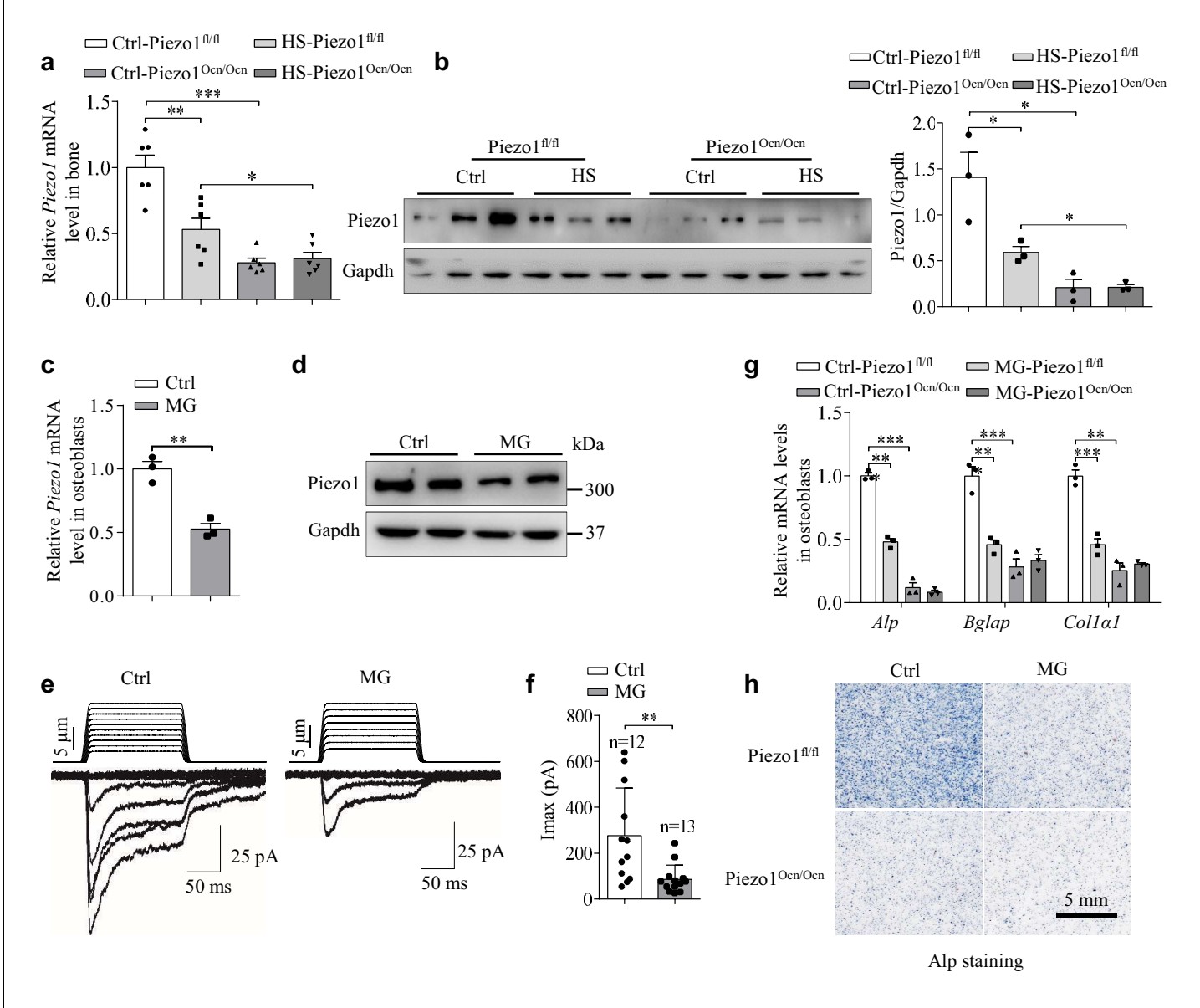

**Figure 5.** Mechanical unloading suppresses Piezo1 expression in Piezo1$^{Ocn/Ocn}$ mice and osteoblasts. (**a, b**) QRT-PCR analysis of *Piezo1* mRNA level (**a**) and western blot analysis of Piezo1 protein level (**b**) in bone tissues from the groups of mice with the indicated treatment conditions. (**c, d**) QRT-PCR analysis of *Piezo1* mRNA level and western blot analysis of Piezo1 protein level in osteoblasts under control (Ctrl) and simulated microgravity (MG) conditions. (**e**) Representative traces of poking-induced inward currents recorded at −60 mV in osteoblasts under Ctrl and MG conditions. (**f**) Scatter plots of the maximal poking-induced currents in osteoblasts under Ctrl and MG conditions. (**g**) QRT-PCR analysis of *Alp*, *Bglap* and *Col1α1* mRNA levels in primary osteoblasts isolated from Piezo1$^{fl/fl}$ and Piezo1$^{Ocn/Ocn}$ mice under Ctrl and MG conditions. (**h**) Representative images of Alp staining in osteoblasts isolated from the indicated mice under Ctrl and MG conditions. Scale bar: 5 mm. All data are the mean ± s.e.m. from three independent experiments. **, p<0.01; ***, p<0.001.

DOI: https://doi.org/10.7554/eLife.47454.016

The following source data is available for figure 5:

**Source data 1.** The data and statistical analysis of the effect of mechanical unloading on Piezo1 expression in osteoblasts.
DOI: https://doi.org/10.7554/eLife.47454.017

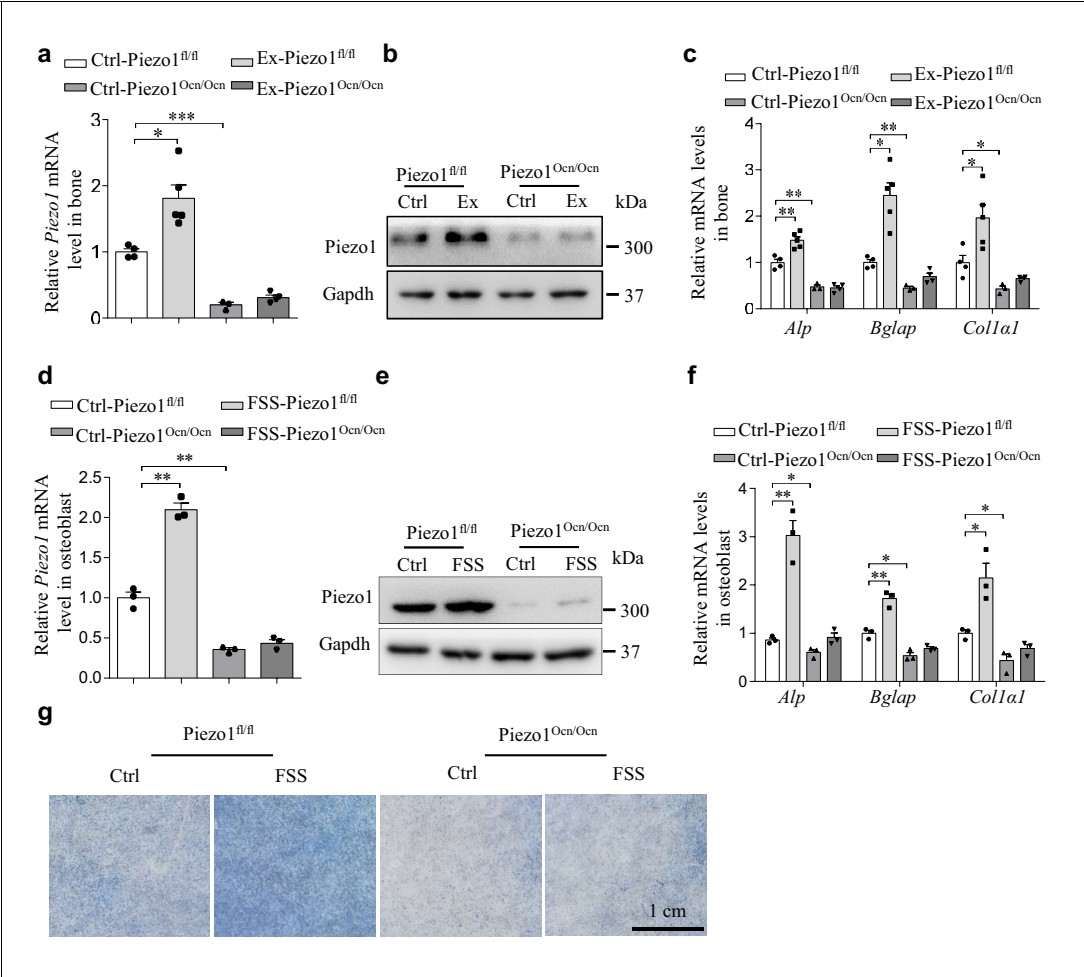

**Figure 6.** Mechanical loading promotes Piezo1 expression in osteoblasts. (**a, b**) QRT-PCR analysis of *Piezo1* mRNA level (**a**) and Western blotting analysis of Piezo1 protein level (**b**) in bone tissues from the groups of mice with the indicated treatment conditions. Ex, Exercise. Ctrl-Piezo1[fl/fl] group, n = 4; Ex-Piezo1[fl/fl] group, n = 5; Ctrl-Piezo1[Ocn/Ocn] group, n = 3; Ex-Piezo1[Ocn/Ocn] group, n = 4. (**c**) QRT-PCR analysis of *Alp*, *Bglap* and *Col1α1* mRNA levels in bone tissues collected from the groups of mice as indicated above. (**d, e**) QRT-PCR analysis of *Piezo1* mRNA level (**d**) and Western blotting analysis of Piezo1 protein level (**e**) in osteoblasts isolated from Piezo1[fl/fl] and Piezo1[Ocn/Ocn] mice under control (Ctrl) and fluid shear stress (FSS) conditions. (**f**) QRT-PCR analysis of *Alp*, *Bglap* and *Col1α1* mRNA levels in osteoblasts isolated from the indicated mice under Ctrl and FSS conditions. (**g**) Representative images of Alp staining in osteoblasts isolated from the indicated mice under Ctrl and FSS conditions. Scale bar: 1 cm. All data are the mean ± s.e.m. from three independent experiments. *, $p < 0.05$; **, $p < 0.01$; ***, $p < 0.001$.

DOI: https://doi.org/10.7554/eLife.47454.018

The following source data is available for figure 6:

**Source data 1.** The data and statistical analysis of the effect of mechanicalloading on Piezo1 expression in osteoblasts.
DOI: https://doi.org/10.7554/eLife.47454.019

osteoblasts might constantly experience mechanical forces such as shear stress and gravity changes. Here, we have found that Piezo1 is expressed in primary osteoblasts and osteocytes, mediates mechanically activated cationic currents and Yoda1 (a Piezo1 chemical activator)-induced $Ca^{2+}$ influx, and controls the osteogenesis process via downstream $Ca^{2+}$ signaling pathways (*Figure 2*). Thus, Piezo1 plays an important role in converting mechanical stimuli into osteogenesis of osteoblasts.

Strikingly, either loss of Piezo1 in the Piezo1[Ocn/Ocn] mice or HS-treatment of the Piezo1[fl/fl] mice led to severely impaired osteogenesis and bone formation, integrity and strength (*Figures 3* and *4*), demonstrating the reciprocal relationship between mechanical loading and the Piezo1 channel in determining the mechanotransduction process during bone formation and remodeling. A further highlight of this relationship is the influence of mechanical loading on the expression level of Piezo1 (*Figure 5* and *Figure 6*). Both microgravity treatment of osteoblasts and mechanical unloading of

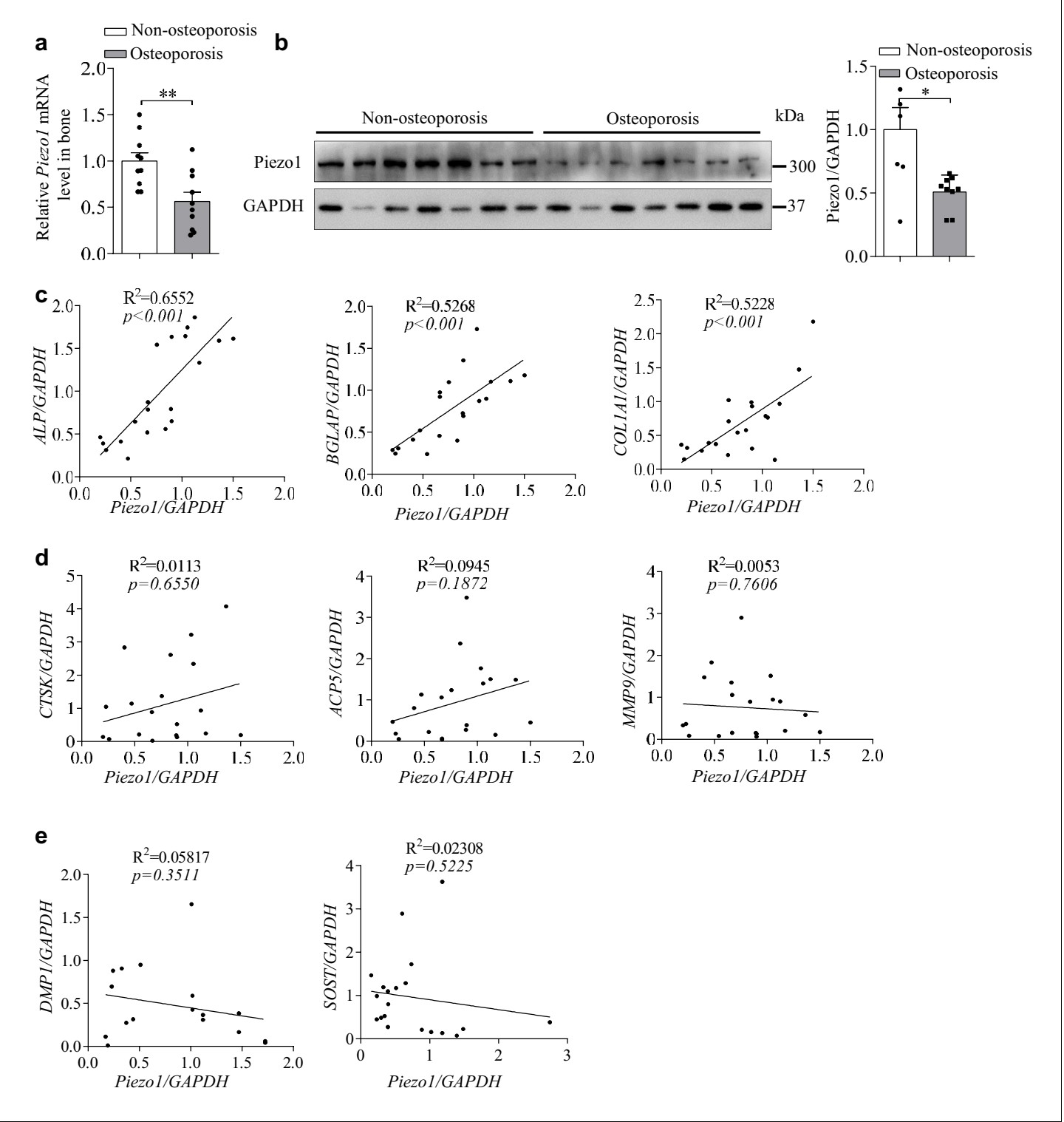

**Figure 7.** Osteoporosis bone specimens have decreased Piezo1 expression and correlated defective osteoblast function. (a) QRT-PCR analysis of *Piezo1* mRNA level in bone specimens from two T-score groups. T > −2.5 group, n = 10, and T ≤ −2.5 group, n = 10. (b) Western blot analysis of Piezo1 protein level in bone specimens from two T-score groups. T > −2.5 group, n = 7, and T≤- 2.5 group, n = 7. Quantification of Piezo1 protein level was normalized to GAPDH. (c) Correlation analysis between *Piezo1* level and the levels of *ALP, BGLAP* or *COL1α1*. T > −2.5 group, n = 10, and T ≤ −2.5 group, n = 10. (d) Correlation analysis between *Piezo1* levels and the levels of *CTSK* (cathepsin K), *ACP5* (acid phosphatase 5) or *MMP9* (matrix metallopeptidase 9). T > −2.5 group, n = 10, and T ≤ −2.5 group, n = 10. (e) Correlation analysis between *Piezo1* levels and the level of *DMP1* (dentin matrix acidic phosphoprotein 1 ) or *SOST* (sclerostin). All data are the mean ± s.e.m. *, p<0.05; **, p<0.01; ***, p<0.001.
*Figure 7 continued on next page*

*Figure 7 continued*

DOI: https://doi.org/10.7554/eLife.47454.020

The following source data is available for figure 7:

**Source data 1.** The data and statistical analysis of the relationship between Piezo1 expression levels and bone formation in human specimens.
DOI: https://doi.org/10.7554/eLife.47454.021

the mice reduced the expression level of Piezo1. By contrast, fluid shear stress treatment osteoblasts and mechanical loading of mice increased the expression level of Piezo1 (*Figure 5* and *Figure 6*). Consistently, previous studies have shown that hydrostatic pressure enhanced the expression of Piezo1 in mesenchymal stem cells and promoted osteogenesis (*Sugimoto et al., 2017*). Furthermore, a stiffer mechanical microenvironment increased the expression of Piezo1 in glioma cells, which in turn enhanced tissue stiffness (*Chen et al., 2018*). Collectively, these data are consistent with a positive feedback loop between the Piezo1 mechanosensor and the mechanical loading experienced by the mechanosensitive cells and organs.

Bone is highly sensitive to changes of daily mechanical loading and gravity. It has been documented that bone mineral density decreased at 1% per month at the lumbar spine and 1–1.6% per month at the hip in the crew members of the international space station (*LeBlanc et al., 2000*; *Vico and Hargens, 2018*). Thus, bone loss is one of the most serious problems induced by long-term weightlessness during space flight or in bedridden individuals. The revelation of the positive feedback relationship between Piezo1 and mechanical loading in bone remodeling provides a mechanistic explanation for mechanical-unloading-induced bone loss. On the basis of the results from the simulated microgravity experiments (*Figure 5*), the lack of gravitational force or mechanical loading during long-term spaceflight or in bedridden individuals might decrease the expression and function of Piezo1 in osteoblasts, which in turn leads to impaired osteogenesis and bone loss. In line with this, we found an apparent correlation between the expression level of Piezo1 and the bone loss in osteoporosis patients (*Figure 7*). Importantly, both the mechanical-unloading-induced bone loss and the mechanical-loading-induced osteogenesis were blunted in Piezo1$^{Ocn/Ocn}$ mice (*Figure 4*), suggesting a causal involvement of Piezo1 in bone remodeling. Thus, activating Piezo1 in the bone might represent a novel strategy for preventing or treating mechanical-unloading-induced bone loss. The success in identifying Piezo1 chemical activators such as Yoda1 (*Syeda et al., 2015*), Jedi1 and Jedi2 (*Wang et al., 2018*) appears to show the promise to be fulfilled in developing Piezo1-based therapeutics.

Given that the Ocn-Cre mice used in the study could also drive Cre expression in osteocytes (*Zhang et al., 2002*), which are derived from osteoblasts and also considered to be mechanosensitive cells in the bone, the observed defects in the bone formation of the Piezo1$^{Ocn/Ocn}$ mice could also be contributed by the expression of Piezo1 in osteocytes. Indeed, we have found that Piezo1 also functions in the osteocytes (*Figure 2—figure supplement 2*). The specific contribution of osteocyte-expressing Piezo1 in bone formation remains to be determined by using osteocyte-specific Cre lines in future studies. Nevertheless, we have demonstrated that Piezo1 plays a critical role in controlling the formation and mechanical-loading-dependent remodeling of the bone in mouse models and it is closely related with the occurrence of osteoporosis in human patients.

## Materials and methods

### Key resources table

| Reagent type (species) or resource | Designation | Source or reference | Identifiers | Additional information |
|---|---|---|---|---|
| Genetic reagent (*M. musculus*) | Ocn-Cre | PMID:27966526 | RRID:MGI:5514364 | Dr. Xiaochun Bai (Southern Medical University, Guangzhou, China) |

*Continued on next page*

*Continued*

| Reagent type (species) or resource | Designation | Source or reference | Identifiers | Additional information |
|---|---|---|---|---|
| Genetic reagent (*M. musculus*) | Piezo1$^{fl/fl}$ | PMID:26001274 | RRID:MGI:3603204 | Dr. Ardem Patapoutian (The Scripps Research Institute, La Jolla, United States) |
| Cell line (*M. musculus*) | MC3T3-E1 | China Infrastructure of Cell Line Resource | RRID:CVCL_0409 | |
| Cell line (*M. musculus*) | RAW264.7 | China Infrastructure of Cell Line Resource | RRID:CVCL_0493 | |
| Antibody | Rabbit anti-Piezo1 | Proteintech | Cat#:15939–1-AP | WB (1:1000) |
| Antibody | Mouse anti-Gapdh | ZSGB-BIO | Cat#:TA-08; RRID:AB_2747414 | WB (1:1000) |
| Antibody | Rabbit anti-p-CamkII | Cell Signaling Technology | Cat#:12716 s | WB (1:1000) |
| Antibody | Rabbit anti-CamkII | Cell Signaling Technology | Cat#:4436 s | WB (1:1000) |
| Antibody | Rabbit anti-p-Creb | Cell Signaling Technology | Cat#:9198 s | WB (1:1000) |
| Antibody | Rabbit anti-Creb | Cell Signaling Technology | Cat#:9197 s | WB (1:1000) |
| Antibody | Rabbit anti-Runx2 | Cell Signaling Technology | Cat#:12556 s | WB (1:1000) |
| Antibody | Rabbit anti-Atf4 | Cell Signaling Technology | Cat#:11815 s | WB (1:1000) |
| Antibody | Rabbit anti-Col1α1 | Abcam | Cat#:ab64883 | IHC (1:400) |
| Antibody | Rabbit anti-Ocn | Proteintech | Cat#:23418–1-AP | IHC (1:200) |
| Recombinant DNA reagent | pCAG-Cre-IRES2-GFP | Shanghai Biological Technology Co., Ltd. enzyme research | Cat#:MY8104 | |
| Recombinant DNA reagent | pIRES-EGFP | Addgene | Addgene plasmid Cat#:45567 | |
| Sequence-based reagent | RT-qPCR primers | This paper | | See *Supplementary file 2* |
| Sequence-based reagent | siRNA | This paper | | See 'Materials and methods' |
| Peptide, recombinant protein | Recombinant mouse RANKL protein | R and D | 462-TR-010 | |
| Commercial assay or kit | PrimeScript RT reagent Kit | TAKARA | RR037A | |
| Commercial assay or kit | TB Green Premix Ex Taq II (Tli RNaseH Plus) | TAKARA | RR820A | |
| Commercial assay or kit | Vector Blue Substrate kit | Vector Laboratories | SK-5300 | |
| Commercial assay or kit | DAB kit | ZSGB-BIO | ZLI-9017 | |
| Commercial assay or kit | Mouse PINP ELISA Kit | Immunoway | KE1744 | |
| Commercial assay or kit | Mouse Ocn ELISA Kit | NOVUS | NBP2-68151 | |
| Commercial assay or kit | Mouse CTX-I ELISA Kit | Sangon Biotech | D720090 | |

*Continued on next page*

*Continued*

| Reagent type (species) or resource | Designation | Source or reference | Identifiers | Additional information |
|---|---|---|---|---|
| Chemical compound, drug | Dulbecco's Modified Eagle Medium (DMEM) | Gibco/Thermo Fisher | Gibco/Thermo Fisher:11965118 | |
| Chemical compound, drug | Minimum Essential Medium (MEM) α | HyClone | SH30265.01B | |
| Chemical compound, drug | Opti-MEM I Reduced Serum Medium | Thermo Fisher | Cat#:31985070 | |
| Chemical compound, drug | Gibco Fetal Bovine Serum | Gibco/Thermo Fisher | Gibco/Thermo Fisher:10099141C | |
| Chemical compound, drug | Penicillin/ streptomycin | Thermo Fisher | Cat#:15140122 | |
| Chemical compound, drug | Dexamethasone | Sigma | Sigma D1796 | |
| Chemical compound, drug | Ascorbic acid | Sigma | Sigma A4544 | |
| Chemical compound, drug | β-glycerophosphate | Sigma | Sigma G9422 | |
| Chemical compound, drug | Collagenase type 2 | Worthington | Cat#:LS004176 | |
| Chemical compound, drug | EDTA | Amresco | Cat#:0322 | |
| Chemical compound, drug | Lipofectamine RNAiMAX | Thermo Fisher | Cat#:13778150 | |
| Chemical compound, drug | Lipofectamine 3000 | Thermo Fisher | Cat#:L3000015 | |
| Chemical compound, drug | Alcian Blue | Sigma | Cat#:A5268 | |
| Chemical compound, drug | Alizarin red | Sigma | Cat#:A5533 | |
| Chemical compound, drug | Calcein green | Sigma | Cat#:C0875 | |
| Software, algorithm | GraphPad Prism | GraphPad Prism (https://www.graphpad.com) | RRID:SCR_015807 | Version 6 |
| Software, algorithm | Osteomeasure Analysis System | | OM-HRDVS | |
| Software, algorithm | Adobe Illustrator | Adobe Illustrator (http://www.adobe.com) | RRID:SCR_010279 | |
| Software, algorithm | ImageJ | ImageJ (http://imagej.nih.gov/ij/) | RRID:SCR_003070 | |

## Animals

To delete Piezo1 specifically in osteoblasts, the conditional KO mice were generated by crossing the Piezo1 floxed mice (Piezo1$^{fl/fl}$) (a generous gift from Dr. Ardem Patapoutian)(*Cahalan et al., 2015*) with the Ocn-Cre transgenic mice (a generous gift from Dr. Xiaochun Bai) (*Huang et al., 2016*). We selected Piezo1$^{Ocn/Ocn}$ as experimental mice, Piezo1$^{fl/fl}$ littermates served as controls. The newborn mice were analyzed by polymerase chain reaction genotyping using genomic DNA from the tail. All animal studies were performed according to approved guidelines for the use and care of live animals (Guideline on Administration of Laboratory Animals released in 1988, and 2006 Guideline on Humane Treatment of Laboratory Animals from China). All the experimental procedures were approved by the Committees of Animal Ethics and Experimental Safety of the China Astronaut Research and Training Center (Reference number: ACC-IACUC-2017–003).

## Cell culture and differentiation

The mouse osteoblast cell line MC3T3-E1 was maintained in minimum essential Eagle's medium, α modification (α-MEM) (Gibco) containing 10% fetal bovine serum (Gibco), 100 units/ml penicillin G, and 100 μg / ml streptomycin (Gibco) at 37°C with 5% $CO_2$. Osteogenic medium was prepared by supplementing the maintenance medium with 10 nM dexamethasone (Sigma), 50 μg/ml of ascorbic acid (Sigma) and 10 mM β-glycerophosphate (Sigma). The murine osteoclast cell line RAW 264.7 was maintained in Dulbecco's modified Eagle's medium (DMEM) with 10% fetal bovine serum, 100 units/ml penicillin G, and 100 μg / ml streptomycin.

Osteoclast-induced medium was prepared by supplementing the maintenance medium with 50 ng/ml recombinant mouse RANKL protein (R and D). All of the cell lines that were utilized were *Mycoplasma*-free, as determined by Q-PCR analyses. Cell line identity was validated by the vendors.

Primary osteoblasts were isolated from the calvarial bone of newborn (1–3 d) mice by enzymatic digestion in α-MEM with 0.1% collagenase and 0.2% dispase as described (*Zhao et al., 2018*), and were cultured in α-MEM with 10% FBS. After 2 days, cells were reseeded and cultured in osteogenic medium for the osteoblast differentiation assay.

Primary osteocytes were obtained after removal of the bone marrow, and then sequential collagenase and EDTA digestions of the long bone (*Stern et al., 2012*). Tibias and femurs of 7–9-week-old mice were cleaned to remove muscle and connective tissue. Epiphyses were cut, bone marrow was flushed and the bone was cut into 1-mm to 2-mm lengths. These fragments were incubated in 1 mg/ml collagenase solution for 30 min at 37°C and this cell suspension was discarded. This was repeated two more times, for a total of three digestions. The remaining bone fragments were washed with PBS and incubated for 30 min at 37°C with EDTA (5 mM, PH 7.4) in PBS. Cell suspension was again discarded, and bone fragments were washed with PBS. This was repeated two more times for the incubation of bone chips with collagenase and EDTA. Cell suspension was again discarded and the bone fragments were finally incubated with 1 mg/ml collagenase for 30 min at 37°C. Cells were collected, passed through a 70-μm nylon mesh and washed twice. These cells were used for subsequent testing.

## Cell transfection

Cells for RNA interference were transfected with *Piezo1* siRNA or NC at 70% confluence using Lipofectamine RNAiMAX in OptiMEM as per the manufacturer's instructions (Invitrogen). Sequences of the siRNA probes were as follows: Piezo1 negative control siRNA (NC), 5′-UUCUCCGAACGUG UCACGUTT-3′; Piezo1 siRNA, 5′-CACCGGCATCTACG TCAAATA-3′.

pIRES-EGFP and pCAG-Cre-IRES2-GFP plasmid transfection, cells at 70% confluence were transfected with 500 ng pIRES-EGFP (Ctrl) and pCAG-Cre-IRES2-GFP (Cre) plasmid using Lipofectamine 3000 in OptiMEM as per the manufacturer's instructions (Invitrogen), cells were analyzed for QRT-PCR, Western Blot and whole cell electrophysiology mechanical stimulation after transfection for 48 hr. Later, cells were analyzed for alp staining after transfection for 5 days and for Alizarin red staining after transfection for 14 days.

## Whole-cell electrophysiology after mechanical stimulation

MC3T3-E1, RAW 264.7, primary osteoblasts or osteocytes isolated from Piezo1[fl/fl] and Piezo1[Ocn/Ocn] mice were cultured on 5 cm$^2$ coverslips. For pCAG-Cre-IRES2-GFP-mediated knockout experiments, the primary osteoblasts derived from Piezo1[fl/fl] mice were transfected with pIRES-EGFP and pCAG-Cre-IRES2-GFP plasmid for 48 hr, and cells with green fluorescence were isolated for electrophysiological recording. For simulated microgravity treatment, cells were cultured on coverslips in a six-well plate for 24 hr. Next, the coverslips were transferred to a cell cabinet, which was filled up with medium and sealed to prevent air bubbles. The cell cabinet was incubated in the clinostat system, with the clinostat being continuously rotated at 30 rpm/min, at 37°C for 16 hr. The control group was cultured in the same manner without clinorotation. Cells that were grown on coverslips were directly subjected to electrophysiological recording. All experiments were performed at room temperature (22–25°C). Mechanical stimulation was delivered to the cell during the patch-clamp recording at an angle of 80° using a fire-polished glass pipette (tip diameter 3–4 μm) as previously described (*Zhao et al., 2016*). The downward movement of the probe toward the cell was driven by a Clampex controlled piezo-electric crystal micro-stage (E625 LVPZT Controller/Amplifier; Physik

Instrument). The probe had a velocity of 1 μm/ms during the downward and upward motion, and the stimulus was maintained for 150 ms. A series of mechanical steps in 1 μm increments was applied every 20 s, and the currents were recorded at a holding potential of −60 mV. GsMTx4 (Tocris Bioscience) at a concentration of 4 μM was added to the recording chamber for 30 min before the recording. For siRNA-mediated knockdown experiments, the MC3T3-E1 cells were transfected with 100 nM siRNA using Lipofectamine 3000 (Invitrogen) following the manufacturer's instructions. After 4 to 6 hr, the medium was replaced with fresh α-MEM medium. 48 hr after transfection, the cells were dissociated with trypsin EDTA (GIBCO) and triturated in α-MEM medium (Life Technologies) supplemented with 10% bovine serum. The resulting cells were placed on coverslips and cultured in a humidified incubator in 5% $CO_2$ at 37°C for 4 hr before the start of recording.

## Single-cell Fura-2 $Ca^{2+}$ imaging

Primary osteoblasts derived from WT and Piezo1 cKO mice and grown on coverslips were subjected to single-cell Fura-2 $Ca^{2+}$ imaging as described previously (*Wang et al., 2018*). The Yoda1-induced amplitude change of the 340/380 fluorescence ratio was calculated by subtracting the baseline ratio prior to Yoda1 application. Yoda1 was solubilized in DMSO as a stock solution of 30 mM and diluted to a final concentration of 30 μM using the $Ca^{2+}$ imaging buffer.

## Quantitative RT-PCR (QRT-PCR)

Total RNA from bone tissues or cells was extracted with TRIzol Reagent (Invitrogen) according to the manufacturer's instructions. RNA (0.5 μg) was reverse transcribed with a PrimeScript RT reagent kit (TaKaRa) according to the manufacturer's instructions. cDNA were used to detect mRNA expression by quantitative PCR using TB Green Premix Ex Taq II (Tli RNaseH Plus) (TaKaRa). *Gapdh* was used as a normal control for mRNA. Primers used are listed in *Supplementary file 2*.

## Western blot analysis

Cells were lysed in lysis buffer (50 mM Tris, pH 7.5, 250 mM NaCl, 0.1% SDS, 2 mM dithiothreitol (DTT), 0.5% NP-40, 1 mM PMSF and protease inhibitor cocktail) on ice for 30 min. Bone tissues were ground with a mortar in liquid nitrogen and were lysed in lysis buffer at 4°C for 30 min. Protein fractions were collected by centrifugation at 12,000 g, 4°C for 10 min and then 10 μg of lysates were subjected to SDS-PAGE and transferred to polyvinylidene difluoride (PVDF) membranes. The membranes were blocked with 5% skimmed milk and incubated with specific antibodies overnight. We used the following antibodies to examine the protein levels in the lysates: Piezo1 (Proteintech, Cat# 15939–1-AP), p-CamkII (CST, Cat# 12716 s), CamkII (CST, Cat# 4436 s), p-Creb (CST, Cat# 9198 s), Creb (CST, Cat# 9197 s), Runx2 (CST, Cat# 12556 s), ATF4 (CST, Cat# 11815 s) and Gapdh (ZSGB-BIO, Cat# TA-08) . The ratios of the protein-band intensities relative to that of Gapdh were calculated for each sample using Image J.

## Alkaline phosphatase staining

Alkaline phosphatase staining was monitored using a Vector Blue substrate kit (procedure number SK-5300, Vector Laboratories). According to the protocol, MC3T3-E1 or primary osteoblasts were incubated with the substrate working solution for 20–30 min. The whole procedure was protected from light.

## Alizarin red staining

Cells were fixed in 4% paraformaldehyde for 5 min and rinsed with double-distilled $H_2O$. Cells were stained with 40 mM Alizarin red S (Sigma), pH 4.0, for 30 min with gentle agitation. Cells were rinsed five times with double-distilled $H_2O$ and then rinsed for 15 min using 1 × PBS with gently agitation.

## Skeletal whole-mount staining

For skeletal preparation, whole-mount skeletal preparations of 5–7-day-old Piezo1[fl/fl] and Piezo1[Ocn/Ocn] mice were prepared by removing the skin and internal organs of the mice before immersion in 95% ethanol for 1–3 days. Specimens were stained with 0.015% Alcian Blue (Sigma) in 80% ethanol with 20% acetic acid. After staining, specimens were washed twice in 95% alcohol for 2 hr, cleared in

1% KOH for 5 hr and stained in 0.005% Alizarin red (Sigma) in 1% KOH for 1 hr. They were then cleared through 20%, 50%, and 80% glycerine in 1% KOH, then stored in 100% glycerine.

## MicroCT analysis

For the distal femur, the whole secondary spongiosa of the left distal femur from each mouse was scanned ex vivo using a microCT system (μCT40, SCANCO MEDICAL, Switzerland). Briefly, 640 slices with a voxel size of 10 μm were scanned in the region of the distal femur beginning at the growth plate and extending proximally along the femur diaphysis. Eighty continuous slices beginning at 0.1 mm from the most proximal aspect of the growth plate in which both condyles were no longer visible were selected for analysis. Cortical bone measurements were performed in the diaphyseal region of the femur starting at a distance of 5.0 mm from the distal growth plate and extending a further longitudinal distance of 80 slices in the proximal direction.

## Three-point bending analysis

Immediately after the dissection, the femurs were stored in 70% ethanol. Before mechanical testing, the bones were rinsed in PBS. The three-point bending test (span length, 4.0 mm; loading speed 0.50 mm/s) at the mid femur was made using Texture Analyzer Texture Pro CT V1.6 Build, Brookfield Engineering Labs Inc.

## Assessment of new bone formation

The mice were injected intraperitoneally with calcein green (10 mg/kg body weight) in a time sequence of 10 d and 2 d before euthanasia. The tibias were harvested for undecalcified histology analysis. Unstained 15 μm sections were examined using fluorescence microscopy. Statistical analyses were performed with the Osteomeasure Analysis System.

## Immunohistochemistry

The tibias of mice were fixed with 4% buffered formalin and embedded with paraffin after decalcification with 10% EDTA for 10–15 days, and 5–7 μm sections were prepared on a rotation microtome. Paraffin-embedded sections were deparaffinized in xylene, and rehydrated. Antigen retrieval was performed in citrate buffer (pH 6.0) for 15 min at 94–96°C. The sections were incubated with 5% goat serum for 1 hr at room temperature. The samples were stained with Col1α1 (abcam, Cat# ab64883) and Ocn (proteintech, Cat# 23418–1-AP) antibody overnight at 4°C. After three washes in PBS, corresponding biotinylated secondary antibodies were then added and incubated for 30 min at room temperature. Negative-control experiments were carried out by omitting the primary antibodies. DAB (ZSGB-bio) was used as chromogen, and hematoxylin was used to counterstain. Statistical analyses were performed with the Osteomeasure Analysis System.

## Serum analysis

The analyses were performed according to the manufacturer's instructions for serum concentrations of PINP (ELISA, Immunoway), Ocn (ELISA, NOVUS) and CTX-1 (ELISA, Sangon Biotech). In brief, 50 μl of serum was pipetted in duplicate into the wells of the precoated ELISA plate. Then, 50 μl of antibody solution was added to each well and incubated at room temperature for 90 min on a shaking device. After incubation, the plates were washed three times with wash buffer.

## Hind-limb suspension mouse model

The hindlimb-unloading procedure was achieved by tail suspension. Briefly, the 3 month Piezo1[fl/fl] or Piezo1[Ocn/Ocn] mice were individually caged and suspended by the tail using a strip of adhesive surgical tape attached to a chain hanging from a pulley. The mice were suspended at a 30° angle to the floor with only the forelimbs touching the floor; this allowed the mice to move and to access food and water freely. The mice were subjected to hindlimb unloading through tail suspension for 28 d. After euthanasia, the whole bone tissues were collected. All animal studies were performed according to approved guidelines for the use and care of live animals (Guideline on Administration of Laboratory Animals released in 1988, and 2006 Guideline on Humane Treatment of Laboratory Animals from China). All of the experimental procedures were approved by the Committees of Animal Ethics

and Experimental Safety of the China Astronaut Research and Training Center (Reference number: ACC-IACUC-2017–003).

## Microgravity simulation

To simulate microgravity, we used a 2D clinostat, which was designed and provided by the China Astronaut Research and Training Center (Beijing, China). Rotation causes a gravity vector that is not recognizable by cells. Therefore, the device prevents the cells from feeling gravity. In the present study, cells were seeded at a density of $1 \times 10^6$ cells in 25 cm$^2$ cell-culture flasks or $2 \times 10^5$ cells on 5 cm$^2$ coverslips adhered to the flask. After cell adhesion, flasks or flasks mounted with coverslips were filled up with culture medium to prevent the presence of air bubbles. The dishes were fixed carefully to the rotating panel of the clinostat system, and rotated at a constant speed of 30 rpm/min for 24 hr to simulate microgravity (0.01 g). For the control, cells were cultured in the same chamber mounted on a stationary clinostat (1 g).

## Exercise mouse model

At 4 months of age, mice were divided into four groups: Ctrl- Piezo1$^{fl/fl}$, Exercise (Ex)-Piezo1$^{fl/fl}$, Ctrl-Piezo1$^{Ocn/Ocn}$ and Exercise (Ex)-Piezo1$^{Ocn/Ocn}$. Each exercise group was subjected to running on a treadmill (Zhishuduobao, Beijing, China) at a 5° incline and speed of 12 m/min, 30 min/day for 21 consecutive days. All animal studies were performed according to approved guidelines for the use and care of live animals (Guideline on Administration of Laboratory Animals released in 1988, and 2006 Guideline on Humane Treatment of Laboratory Animals from China). All of the experimental procedures were approved by the Committees of Animal Ethics and Experimental Safety of the China Astronaut Research and Training Center (Reference number: ACC-IACUC-2017–003).

## Fluid shear stress experiment

Fluid flow was applied to cells in a parallel plate flow chamber using a closed flow loop. Cells were plated on $22 \times 26$ mm glass cover slips and placed into chambers at 80% confluence. After treatment with 12 dyn/cm$^2$ FSS for 2 hr, the apparatus was maintained at 37 °C throughout the duration of the experiment. The correlation between FSS and flow rate was calculated using the equation: $\tau = 6\mu Q/bh^2$, where Q is the flow rate (cm$^3$/s), $\mu$ is the viscosity of the flow media (0.01 dynes/cm$^2$), h is the height of the channel (0.05 cm), b is the slit width (2.5 cm), and $\tau$ is the wall shear stress (dyne/cm$^2$).

## Preparation of human bone tissue

The bone tissues of 10 osteoporotic patients and 10 non-osteoporotic people were collected from a clinical setting. The patients were recruited at between 65 and 90 years of age. The classification of the patients into the osteoporotic and non-osteoporotic groups was based on DXA evaluation. We measured the T-score for BMD in the spine of women. A T-score of −2.5 or lower qualifies as osteoporosis. Others were control patients (T > −2.5). We obtained informed consent from all participants. The study protocol conformed to the ethical guidelines of the 1975 Declaration of Helsinki. All clinical procedures were approved by the Committees of Clinical Ethics in the Second Affiliated Hospital of Soochow University (Reference number: 2016 K-22).

## Statistical analysis

All numerical data are expressed as the mean ± SEM from at least three independent samples. Student's t-test was used for statistical evaluations of two group comparisons. Statistical analysis with more than two groups was performed with one-way analysis of variance (ANOVA). All statistical analyses were performed with Prism software (Graphpad prism for windows, version 6.0). $p < 0.05$ was considered statistically significant.

## Acknowledgements

We would like to thank Prof. Heping Cheng from Peking University for the guidance and careful discussions about the paper, Prof. Xiaochun Bai (Southern Medical University, Guangzhou, China) for providing the Ocn-Cre mice and Prof. Lin Chen (Third Military Medical University, Chongqing, China)

for providing the Osteomeasure Analysis System. This work was supported by the National Natural Science Foundation of China (31825014, 81830061, 31630038, 91740114, 31630090, 81822026, 31800994 and 31700741) to BX, YL SL, WS and YHL, the 1226 project (AWS16J018) to YL, and the Ministry of Science and Technology (2016YFA0500402 and 2015CB910102) to BX.

## Additional information

### Funding

| Funder | Grant reference number | Author |
|---|---|---|
| National Natural Science Foundation of China | 31630038 | Yingxian Li |
| National Natural Science Foundation of China | 91740114 | Yingxian Li |
| National Natural Science Foundation of China | 81830061 | Yingxian Li |
| National Natural Science Foundation of China | 31700741 | Yuheng Li |
| National Natural Science Foundation of China | 31825014 | Bailong Xiao |
| National Natural Science Foundation of China | 31630090 | Bailong Xiao |
| Ministry of Science and Technology of the People's Republic of China | 2016YFA0500402 | Bailong Xiao |
| Ministry of Science and Technology of the People's Republic of China | 2015CB910102 | Bailong Xiao |
| National Natural Science Foundation of China | 31800994 | Weijia Sun |
| National Natural Science Foundation of China | 81822026 | Shukuan Ling |
| 1226 Project | AWS16J018 | Yingxian Li |

The funders had no role in study design, data collection and interpretation, or the decision to submit the work for publication.

### Author contributions

Weijia Sun, Resources, Data curation, Formal analysis, Validation, Investigation, Methodology, Writing—original draft; Shaopeng Chi, Resources, Data curation, Formal analysis, Validation, Investigation, Methodology; Yuheng Li, Data curation, Formal analysis, Visualization, Methodology; Shukuan Ling, Conceptualization, Supervision, Visualization; Yingjun Tan, Resources, Methodology; Youjia Xu, Resources; Fan Jiang, Guohui Zhong, Dengchao Cao, Xiaoyan Jin, Dingsheng Zhao, Xingcheng Gao, Zizhong Liu, Data curation, Formal analysis; Jianwei Li, Caizhi Liu, Data curation, Formal analysis, Methodology; Bailong Xiao, Conceptualization, Supervision, Funding acquisition, Project administration, Writing—review and editing; Yingxian Li, Conceptualization, Supervision, Funding acquisition, Writing—original draft, Project administration, Writing—review and editing

### Author ORCIDs

Yingxian Li https://orcid.org/0000-0002-5440-3281

### Ethics

Human subjects: The study protocol conformed to the ethical guidelines of the 1975 Declaration of Helsinki. All the clinical procedures were approved by the Committees of Clinical Ethics in the Second Affiliated Hospital of Soochow University (Reference number: 2016-K-22).

Animal experimentation: All animal studies were performed according to approved guidelines for the use and care of live animals (Guideline on Administration of Laboratory Animals released in1988 and 2006 Guideline on Humane Treatment of Laboratory Animals from China). All the experimental procedures were approved by the Committees of Animal Ethics and Experimental Safety of China Astronaut Research and Training Center (Reference number: ACC-IACUC-2017-003).

## Decision letter and Author response

Decision letter https://doi.org/10.7554/eLife.47454.026
Author response https://doi.org/10.7554/eLife.47454.027

## Additional files

### Supplementary files

• Supplementary file 1. Clinical features of fracture patients involved in bone specimens analysis.
DOI: https://doi.org/10.7554/eLife.47454.022

• Supplementary file 2. The sequences of mRNA primers.
DOI: https://doi.org/10.7554/eLife.47454.023

• Transparent reporting form
DOI: https://doi.org/10.7554/eLife.47454.024

### Data availability

All data generated or analysed during this study are included in the manuscript and supporting files.

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
