## [Decision Letter]

Thank you for submitting your article "The mechanosensitive Piezo1 channel is required for bone formation" for consideration by *eLife*. Your article has been reviewed by three peer reviewers, and the evaluation has been overseen by a Reviewing Editor and Harry Dietz as the Senior Editor. The following individuals involved in review of your submission have agreed to reveal their identity: Vanessa Sherk (Reviewer #1) and Nele A Haelterman (Reviewer #2).

The reviewers have discussed the reviews with one another and the Reviewing Editor has drafted this decision to help you prepare a revised submission.

Summary:

In the current manuscript, Sun et al. characterize the contribution of the mechanosensitive channel Piezo1 to bone's ability to respond to mechanical stimuli. The authors find that an Osteocalcin-mediated knockout of Piezo1 renders osteoblast precursors insensitive to mechanical stimuli and severely impairs bone development. In addition, loss of this channel abolishes mechanical-induced bone loss, suggesting that Piezo1 is a key factor in regulating this aspect of bone homeostasis. Overall this paper might break new ground in our understanding of mechanosensing in bone. As noted below, several issues need resolution:

Essential revisions:

1) Although the authors show the effect of Piezo1 deletion in osteoblasts it would be essential to identify it also as being relevant to osteocytes; so this would entail showing that these channels are operative in osteocytes.

2) It would be useful to show that LOADING induces Piezo1 and osteogenesis and that blocking this prevents osteogenesis.

3) There should be more complete phenotyping of the conditional mice including calvariae which are not load sensitive yet may be abnormal; as well as length and morphology.

For details, please see the reviewer comments below.

*Reviewer #1:*

The purpose of this study was to test the role of Piezo1 in conferring mechanosensitivity in osteoblasts using MC3T3 cells, primary osteoblasts from Piezo1Ocn/Ocn and Piezo1fl/fl mice, and human bone samples. Further, experiments were carried out in an osteoclast cell line. The studies were very interesting overall but important questions remain.

1) What was the sex of the animals used for isolating osteoblasts and for the mice euthanized at 2 months? Can the authors demonstrate whether the Piezo1-induces changes in osteoblasts or in the bone phenotype is present in both sexes?

2) The manuscript briefly mentions the possible role of Piezo1 expression in osteocytes in mechanosensing and bone formation. Was there a difference in Sost or DMP1 expression in bone samples between mouse strains? Was Piezo1 expression associated with Sost or DMP1 in the humans samples? Do primary osteocytes express Piezo1?

*Reviewer #2:*

Bone is an active tissue that adapts its strength to changes in mechanical load. Interest in the mechanisms that underlie bone's mechanosensitivity is high, but so far, our knowledge of this aspect of bone biology remains fairly limited.

In the current manuscript, Sun et al. characterize the contribution of the mechanosensitive channel Piezo1 to bone's ability to respond to mechanical stimuli. The authors find that an Osteocalcin-mediated knockout of Piezo1 renders osteoblast precursors insensitive to mechanical stimuli and severely impairs bone development. In addition, loss of this channel abolishes mechanical-induced bone loss, suggesting that Piezo1 is a key factor in regulating this aspect of bone homeostasis. This is a well-written manuscript that presents novel insights in bone homeostasis that are supported by convincing data. In my opinion, it will provide a significant advance to the field after having answered the below questions.

I have no major concerns

*Reviewer #3:*

The authors have clearly shown an effect of deleting Piezo1 in osteoblasts using the Osteocalcin-Cre mice as these mice have reduced bone and shown that reducing Piezo1 in MC3T3 osteoblasts reduces the expression of osteogenic markers. The authors conclude this is due to the channels acting as mechanosensors. Frequently gene or protein expression is equated with a specific function but may actually be working through another mechanism. It would be important to prove lack of mechanosensation is responsible for reduced bone formation.

1) The investigators examined evoked cationic currents with 'mechanical poking' of an osteoblast and an osteoclast cell line. As osteocytes are thought to be more mechanically sensitive than osteoblasts, why not test this bone cell type? Why not also look at the levels of Piezo1 in osteocytes?

2) The authors show that reducing expression of Piezo1 in MC3T3 cells reduced markers of osteogenesis such as alkaline phosphatase, osteocalcin, and collagen type 1. Can any form of mechanical loading such as shear stress or substrate stretching or 'poking induced currents' increase these markers in osteoblasts? An ideal experiment would be to show that 'poking' induced currents induce osteogenesis and this can be blocked by a Piezo1 antagonist.

3) For the mice with the targeted deletion of Piezo1 in osteoblasts, it looks like the skeletons are similar in size. Is this correct? It would be important to show this-for example the bone length, morphology etc. What is the weight of these animals? The calvaria in Figure 3A look very different. Is there a deformity in addition to the open suture? A more careful analysis should be performed.

4) One may not see an effect of unloading on the mice with the targeted deletion of Piezo1 because the bone mass and properties are already at the same level as unloaded wild-type mice. There is a baseline where additional unloading does not result in more bone loss. Therefore, one cannot infer from these studies that these bones are not sensing a lack of loading.

5) In Figure 5 is appears that something upregulated with unloading that is regulating Piezo1 gene expression. This would suggest another mechanosensor, not Piezo1.

I would have thought that it would be more important to show that these channels are responsible for bone formation due to loading instead of focusing on a lack of channels are responsible for bone loss due to unloading.

---

## [Author Response]

Essential revisions:1) Although the authors show the effect of Piezo1 deletion in osteoblasts it would be essential to identify it also as being relevant to osteocytes; so this would entail showing that these channels are operative in osteocytes.

Thanks so much for the suggestion. Osteocytes are also important mechanosensory cells in bone. To explore the role of Piezo1 in osteocytes, we have isolated osteocytes from the bone tissues and examined its mechanical response. Piezo1^fl/fl^ osteocytes displayed poking-induced currents in a step-dependent manner with a maximal current of 64.93 ± 13.69 pA. The osteocytes from Piezo1^Ocn/Ocn^ mice exhibited significantly reduced Piezo1 protein levels and currents (34.22 ± 4.1 pA) (Figure 2—figure supplement 2A-D). In contrast, the expression of osteocyte marker gene Sost was increased in bone tissue from Piezo1^Ocn/Ocn^ mice.

We have included these new data in the revised version of the manuscript and clearly stated in the Discussion section that “Given that the Ocn-Cre mice used in the study could also drive Cre expression in osteocytes (Zhang et al., 2002), which are derived from osteoblasts and also considered as mechanosensitive cells in the bone, the observed defects in bone formation of the Piezo1^Ocn/Ocn^ mice could also be contributed by the expression of Piezo1 in osteocytes. […] Nevertheless, we have demonstrated that Piezo1 plays a critical role in controlling the formation and mechanical loading-dependent remodeling of the bone in mouse models and it is closely related with the occurrence of osteoporosis in human patients.”

2) It would be useful to show that LOADING induces Piezo1 and osteogenesis and that blocking this prevents osteogenesis.

Thanks so much for the suggestion. To further examine whether Piezo1 is involved in loading induced osteogenesis, we used treadmill running model for mouse and fluid shear stress (FSS) treatment for osteoblasts, two commonly used for assessing the role of mechanical loading on skeleton and cell, respectively. The results showed that exercise treatment led to a significant increase in the expression of Piezo1 as well as the differentiation marker genes of osteoblasts in the Piezo1^fl/fl^ mice, but not in the Piezo1^Ocn/Ocn^ mice (Figure 6A-C). Similarly, FSS treatment led to significantly increased expression of Piezo1 and osteoblast marker genes (Figure 6D-F) and Alp activity (Figure 6G) in the osteoblasts isolated from Piezo1^fl/fl^ cells, but not in the Piezo1^Ocn/Ocn^ cells. These results suggest that Piezo1 plays a critical role in mediating loading induced osteogenesis. We have included these exciting new data in the revised manuscript.

3) There should be more complete phenotyping of the conditional mice including calvariae which are not load sensitive yet may be abnormal; as well as length and morphology.

In the revised manuscript, we have provided the whole phenotype for the newborn and adult mice. The newborn Piezo1^Ocn/Ocn^ mice showed incomplete closure of the cranial structure in Piezo1^Ocn/Ocn^ mice (Figure 3A). The 2-month-old Piezo1^Ocn/Ocn^ mice showed shorter stature (Figure 3—figure supplement 1A) and lower body weight (Figure 3—figure supplement 1B). The length of the femur and tibia of the 2-month-old Piezo1^Ocn/Ocn^ mice was apparently shorter than that of the Piezo1^fl/fl^ control mice (Figure 3—figure supplement 1C).

Reviewer #1:

The purpose of this study was to test the role of Piezo1 in conferring mechanosensitivity in osteoblasts using MC3T3 cells, primary osteoblasts from Piezo1Ocn/Ocn and Piezo1fl/fl mice, and human bone samples. Further, experiments were carried out in an osteoclast cell line. The studies were very interesting overall but important questions remain.1) What was the sex of the animals used for isolating osteoblasts and for the mice euthanized at 2 months? Can the authors demonstrate whether the Piezo1-induces changes in osteoblasts or in the bone phenotype is present in both sexes?

Thanks so much for the comments. We apologize for not clearly stating the sex of the mice in the previous version of the manuscript. Male mice were used for isolating osteoblasts and for the mice euthanized at 2 months. We have specified the sex of the mice in the revised manuscript.

We have analyzed the phenotype of female mice. Interestingly, female Piezo1Ocn/Ocn mice show similar defective bone phenotypes as male Piezo1Ocn/Ocn mice (Figure 3—figure supplement 2).

We have included the new data in the revised version of the manuscript.

2) The manuscript briefly mentions the possible role of Piezo1 expression in osteocytes in mechanosensing and bone formation. Was there a difference in Sost or DMP1 expression in bone samples between mouse strains? Was Piezo1 expression associated with Sost or DMP1 in the humans samples? Do primary osteocytes express Piezo1?

Given the reviewer’s comments, we have analyzed the expression and function of Piezo1 in osteocytes. Osteocytes derived from the Piezo1^Ocn/Ocn^ mice had reduced expression of Piezo1 (Figure 2—figure supplement 2A, B). Accordingly, the expression of osteocyte marker gene Sost was increased in bone tissue from Piezo1^Ocn/Ocn^ mice. However, in the human samples, we didn’t find correlation between the expression of Piezo1 and DMP1 and SOST (Figure 7E).

We have included the new data and discussion in the revised version of the manuscript.

Reviewer #3:

The authors have clearly shown an effect of deleting Piezo1 in osteoblasts using the Osteocalcin-Cre mice as these mice have reduced bone and shown that reducing Piezo1 in MC3T3 osteoblasts reduces the expression of osteogenic markers. The authors conclude this is due to the channels acting as mechanosensors. Frequently gene or protein expression is equated with a specific function but may actually be working through another mechanism. It would be important to prove lack of mechanosensation is responsible for reduced bone formation.1) The investigators examined evoked cationic currents with 'mechanical poking' of an osteoblast and an osteoclast cell line. As osteocytes are thought to be more mechanically sensitive than osteoblasts, why not test this bone cell type? Why not also look at the levels of Piezo1 in osteocytes?

Thanks so much for the constructive comment. In light of the reviewer’s comments, we have analyzed the expression and function of Piezo1 in osteocytes. Osteocytes derived from the Piezo1^Ocn/Ocn^ mice had reduced expression of Piezo1 (Figure 2—figure supplement 2A, B). Accordingly, the expression of osteocyte marker gene Sost were increased in bone tissue from Piezo1^Ocn/Ocn^ mice. However, in the human samples, we didn’t find correlation between the expression of PIEZO1 and DMP1 and SOST (Figure 7E).

We have included these new data in the revised version of the manuscript and clearly stated in the Discussion section that “Given that the Ocn-Cre mice used in the study could also drive Cre expression in osteocytes (Zhang et al., 2002), which are derived from osteoblasts and also considered as mechanosensitive cells in the bone, the observed defects in bone formation of the Piezo1^Ocn/Ocn^ mice could also be contributed by the expression of Piezo1 in osteocytes[…] Nevertheless, we have demonstrated that Piezo1 plays a critical role in controlling the formation and mechanical loading-dependent remodeling of the bone in mouse models and it is closely related with the occurrence of osteoporosis in human patients.”

2) The authors show that reducing expression of Piezo1 in MC3T3 cells reduced markers of osteogenesis such as alkaline phosphatase, osteocalcin, and collagen type 1. Can any form of mechanical loading such as shear stress or substrate stretching or 'poking induced currents' increase these markers in osteoblasts? An ideal experiment would be to show that 'poking' induced currents induce osteogenesis and this can be blocked by a Piezo1 antagonist.

In light of the reviewer’s suggestion, we have treated primary osteoblasts with fluid shear stress (FSS). FSS treatment led to significantly increased expression of Piezo1 and osteoblast marker genes (Figure 6D-F) and Alp activity (Figure 6G) in the osteoblasts isolated from Piezo1^fl/fl^ cells, but not in the Piezo1^Ocn/Ocn^ cells. These results suggest that Piezo1 mediated mechanical force induced osteogenesis. Given that we cannot get sufficient amount of the Piezo1 antagonist GsMTX4, we did not performed the blocking experiment as suggested by the reviewer.

3) For the mice with the targeted deletion of Piezo1 in osteoblasts, it looks like the skeletons are similar in size. Is this correct? It would be important to show this-for example the bone length, morphology etc. What is the weight of these animals? The calvaria in Figure 3A look very different. Is there a deformity in addition to the open suture? A more careful analysis should be performed.

The newborn Piezo1^Ocn/Ocn^ mice had similar skeletal size as their Piezo1^fl/fl^ littermates. However, at 8 weeks of age, the male Piezo1^Ocn/Ocn^ mice showed shorter stature (Figure 2—figure supplement 1A) and lower body weight (Figure 2—figure supplement 1B). The length of the femur and tibia of the Piezo1^Ocn/Ocn^ mice was apparently shorter than that of the Piezo1^fl/fl^ control mice (Figure 2—figure supplement 1C). The Piezo1^Ocn/Ocn^ mice exhibited incomplete closure of the cranial structure, but we didn’t find other deformities.

4) One may not see an effect of unloading on the mice with the targeted deletion of Piezo1 because the bone mass and properties are already at the same level as unloaded wild-type mice. There is a baseline where additional unloading does not result in more bone loss. Therefore, one cannot infer from these studies that these bones are not sensing a lack of loading.

In light of the reviewer’s concern, we have used treadmill running model for mouse, which is a common used exercise model to assess the role of mechanical loading on skeleton. Interestingly, we have found that exercise treatment led to a significant increase in expression of Piezo1 and the differentiation marker genes in osteoblasts derived from the Piezo1^fl/fl^ mice, but not from the Piezo1^Ocn/Ocn^ mice (Figure 6A-C). We have included the new data in the revised version of the manuscript.

5) In Figure 5 is appears that something upregulated with unloading that is regulating Piezo1 gene expression. This would suggest another mechanosensor, not Piezo1.

We have described the positive feedback relationship between Piezo1 mechano-sensing and expression in the Discussion section. “Either loss of Piezo1 in the Piezo1^Ocn/Ocn^ mice or HS-treatment of the Piezo1^fl/fl^ mice led to severely impaired osteogenesis and bone formation, integrity and strength (Figure 3 and Figure 4), demonstrating the reciprocal relationship between mechanical loading and the Piezo1 channel in determining the mechanotransduction process during bone formation and remodeling. […] Collectively, these data are consistent with a positive feedback loop between the Piezo1 mechanosensor and the mechanical loading experienced by the mechanosensitive cells and organs.”

We agree with the reviewer how the expression of Piezo1 is regulated remains unclear, which clearly merits future investigations.

I would have thought that it would be more important to show that these channels are responsible for bone formation due to loading instead of focusing on a lack of channels are responsible for bone loss due to unloading.

We thank the reviewer for the constructive comment. In the revised manuscript, we have carried additional experiments to analyze the role of Piezo1 channel in response to the mechanical loading in osteoblasts in vivo and in vitro. We have included the new data in Figure 6 in the revised version of the manuscript.